# How Does Intensive Land Use Affect Low-Carbon Transition in China? New Evidence from the Spatial Econometric Analysis

Xiao Ling [1], Yue Gao [2] and Guoyong Wu [3,*]

1 Business School, Hubei University, Wuhan 430062, China; lingxiao@stu.hubu.edu.cn
2 Business School, Zhengzhou University, Zhengzhou 450001, China; gaoyue980913@163.com
3 Western Modernization Research Institute, Guizhou University, Guiyang 550025, China
* Correspondence: wgyong@126.com

**Abstract:** Anthropogenic land cover change is one of the primary sources of increasing carbon emissions and affects the potential of terrestrial ecosystems to store carbon and act as carbon sinks. As a necessary means to reduce land expansion, land-use intensification significantly impacts greenhouse gas emission reduction and the low-carbon transition of the economy. This paper constructs a framework for the relationship between intensive land use (*ILU*) and low carbon transition (*LCT*), considering direct and spatially driven effects. First, this paper constructs a multidimensional indicator to measure intensive land use and documents the spatial pattern of intensive land use levels in China. Second, this paper assesses the spatial driving effect of intensive land use on China's economic low-carbon transition. Based on data from 283 Chinese cities from 2006–2019 and using a spatial Durbin model, the study provides empirical evidence that intensive land use can significantly promote low-carbon transition in neighboring and economically linked cities (especially in eastern cities, large and medium-sized cities, and veteran economic circles). Tests introducing exogenous policy shocks further confirm the robustness of the findings. In addition, industrial structure transformation and technology spillovers are identified as the dual mechanism channels of intensive land use for low-carbon transition in China, and the spatial driving effect on neighboring cities attenuating with geographic distance is also confirmed.

**Keywords:** intensive land use; low carbon transformation; industrial structure transformation; technology spillovers; national and regional policy; land space planning

## 1. Introduction

Global warming caused by greenhouse gas (GHG) emissions seriously threatens the natural and social environments on which human beings depend for survival [1,2]. The series of chain reactions across ecosystems triggered by greenhouse gases has become a massive challenge for all humanity [3,4]. The International Energy Agency (IEA) estimates that global energy-related carbon dioxide ($CO_2$) emissions will grow by 0.9% in 2022, reaching a record high of over 36.8 Gt [5]. Among them, carbon dioxide emissions from energy combustion and industrial processes account for 89% of total energy-related greenhouse gas emissions; methane from energy combustion, leakage, and venting accounts for 10%. They are all mainly from onshore oil and gas field operations and the production of coal for power. Compared to 1880, 2022 is also the fifth hottest year globally, fraught with extreme weather events [6,7]. As the country with the most rapid economic development in the 20th century, China has become the world's largest emitter of carbon dioxide since 2007, with carbon emissions rising from 8.83 billion tons in 2011 to 9.90 billion tons in 2020 [8]. As the urbanization rate of the population rises (to 64.72% in 2021), large-scale migration and the concentration of human activities will result in continued land expansion and land carbon emissions. China is actively taking on the corresponding obligations to mitigate global warming, noting that it should effectively promote carbon peaking and

carbon-neutral actions. In this context, China urgently needs to find breakthrough solutions to accelerate economic activities' low-carbon transition to control the increasingly severe climate problem [9].

Since the 19th century, land use has influenced terrestrial ecosystem carbon balance through changes in land cover status and the human activities it hosts [10]. It has been recognized as an essential factor influencing regional carbon source/sink patterns [11]. From 1750 to 2011, an estimated 180 Gt has been released globally due to deforestation and other land use changes [12]; more than 66% of energy and 80% of carbon emissions may be related to the scale and productivity of land use [13], and the loss of carbon stocks in terrestrial ecosystems due to the occupation of forest resources worldwide is the second largest source of carbon emissions [14]. The loss of ecosystem carbon stocks will be exacerbated by unintentional land expansion, and the overconcentration of human activities brought about by expansion will also generate high consumption and emissions [15]. As the world's largest carbon emitter, land use carbon emissions have become an essential source of carbon emissions in China, reaching $3.2 \times 10^9$ t in 2015, an increase of about 2.45 times compared with 1999. As of 2020, China's land-use carbon emissions will remain high (see Figure 1). Assuming that the 1.5 °C global temperature control target of the Paris Agreement is to be achieved, further attention needs to be paid to the critical role of intensive land use in the low-carbon transition of the economy. China is implementing policy elements of intensive land utilization to promote a low-carbon transition in economic development. In particular, since the promulgation of the Regulations on the Economical and Intensive Utilization of Land in 2014, land regulation for carbon emission reduction has become an important means of promoting grassroots efforts to achieve carbon neutrality targets. The regulation encourages small-scale centralized and intensive land use and emphasizes green and livable land use [16]. Zhao (2021) assessed the carbon emission reduction contribution of the Outline of China's Overall Land Use Plan (2005–2020). Based on 2005, optimizing the land use structure will contribute 27.6% to the achievement of the target of carbon emission reduction of 40% to 45% per unit of GDP in 2020 [17]. In the context of the country's emphasis on coordinated "economic-ecological" development, intensive land use is crucial for China to achieve the goal of "carbon neutrality" and low-carbon transition.

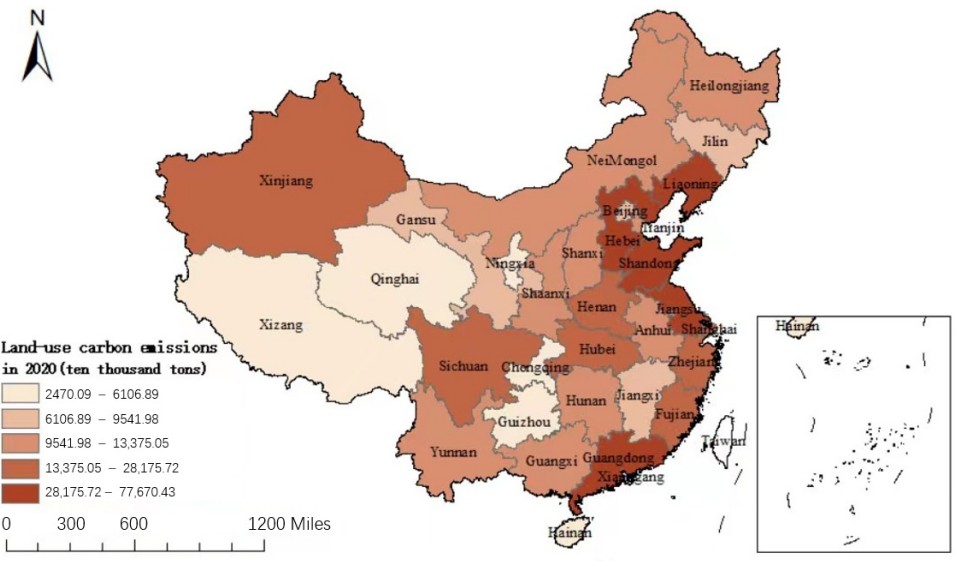

**Figure 1.** Land use carbon emissions for 31 provinces in China in 2020.

In China, fiscal revenue from land concessions has long been local governments' primary income source. To solve the fiscal balance gap, some local governments have been keen to attract industrial investment by taking advantage of their resource endowment

and geographic location [18], dramatically expanding industrial scale while promoting development. This process has not only resulted in massive waste of urban land and rapid urbanization but also led to differences in the spatial pattern of land use carbon emissions [19]. With China's coordinated economic development strategy deepening, the flow of technology, personnel, capital, and other factors between regions has further accelerated. The spatial correlation of economic development, energy consumption, and agricultural activities has broken through the limitations of geographic location. The spatial correlation of land-use carbon emissions will also be further complicated. Under the spatial differences in population distribution and economic resources, the differences in land carbon emissions of each province in China are apparent; carbon emissions are high in the eastern coastal areas (Figure 1). In this context, it is significant to carry out a study on the spatial differences in carbon emissions from land use for the synergistic emission reduction of regional land use.

As land is essential for population, industry, transportation, buildings, and energy use, can improved intensive land use help promote China's low-carbon transition? Moreover, in what ways does it achieve "economic-ecological" synergistic development? Given China's goal of achieving carbon neutrality by 2050, these questions' conceptual and applied consequences are critical to assess. Existing studies have intensely discussed the relationship between land use and carbon emissions. First, some scholars have expressed the hidden concern that land expansion may lead to increased carbon emissions in earlier studies [20,21], arguing that these crude features are an essential cause of higher environmental costs and unsustainable problems [22,23]. Some scholars have also tested the relationship between land use structure and carbon emissions [24]. It is pointed out that the structural imbalance of ecological land, agricultural land, and urban land [25], as well as the increase in the proportion of urban built-up areas [26,27], will lead to a more obvious greenhouse effect. Then, some scholars explored the environmental benefits of optimizing land use. For example, Xie et al. (2018) pointed out that improving industrial land use efficiency at the national level contributes 37.52% to the total $CO_2$ emission reduction, in which R&D investment in intensive land use is the most effective way to promote emission reduction [28]. Additionally, Goh et al. (2018) concluded that decarbonizing land use can be an effective method of reducing carbon emissions [29]. These conclusions are also recognized by scholars such as Peng et al. (2022) [30] and Zhang et al. (2023) [31]. However, some scholars, such as Zhu et al. (2022), believe that China's current land use optimization has caused a greater degree of carbon emissions [32]. Scholars have only analyzed the impact of optimized land use on the intensity and efficiency of carbon emissions in isolation, and their views have not yet reached a consensus. Therefore, despite the intense academic discussion and research on these issues so far, there still exists a vast research space, such as the characteristics of spatial distribution, direct and spillover effects, and impact mechanisms, which constitute the initial motivation of this study.

Based on the consideration of breaking through the limitations of the existing literature, we decided to identify and assess the driving effect of land intensification on China's low-carbon transition from a spatial perspective and evaluate the mechanism of its action in terms of both green upgrading of industries and clean technology spillovers (Figure 2). The possible marginal contribution consists of the following three points. Firstly, we constructed a framework for the relationship between land use intensification and low carbon transition considering both direct and spatial driving effects and numericized land use intensification and low carbon transition in the form of multiple composite indicators. Compared with traditional studies, we provide a more comprehensive analysis from a spatiotemporal perspective (i.e., spatial distribution, spatial autocorrelation, evolution, spillover effects, spatial decay, and spatial heterogeneity). Secondly, compared with the traditional single research method, we adopt the exogenous policy shock test to support the conclusion of the driving effect and examine the dual channels of influence of land intensification and low-carbon transition from the dual perspectives of green transformation of industrial structure and clean technology spillovers, which expands the empirical research in related

fields. Finally, we provide practical policy recommendations for policymakers regarding the efficiency of low-carbon economies and the focus on green, livable, and efficient living and production environments in emerging countries such as China.

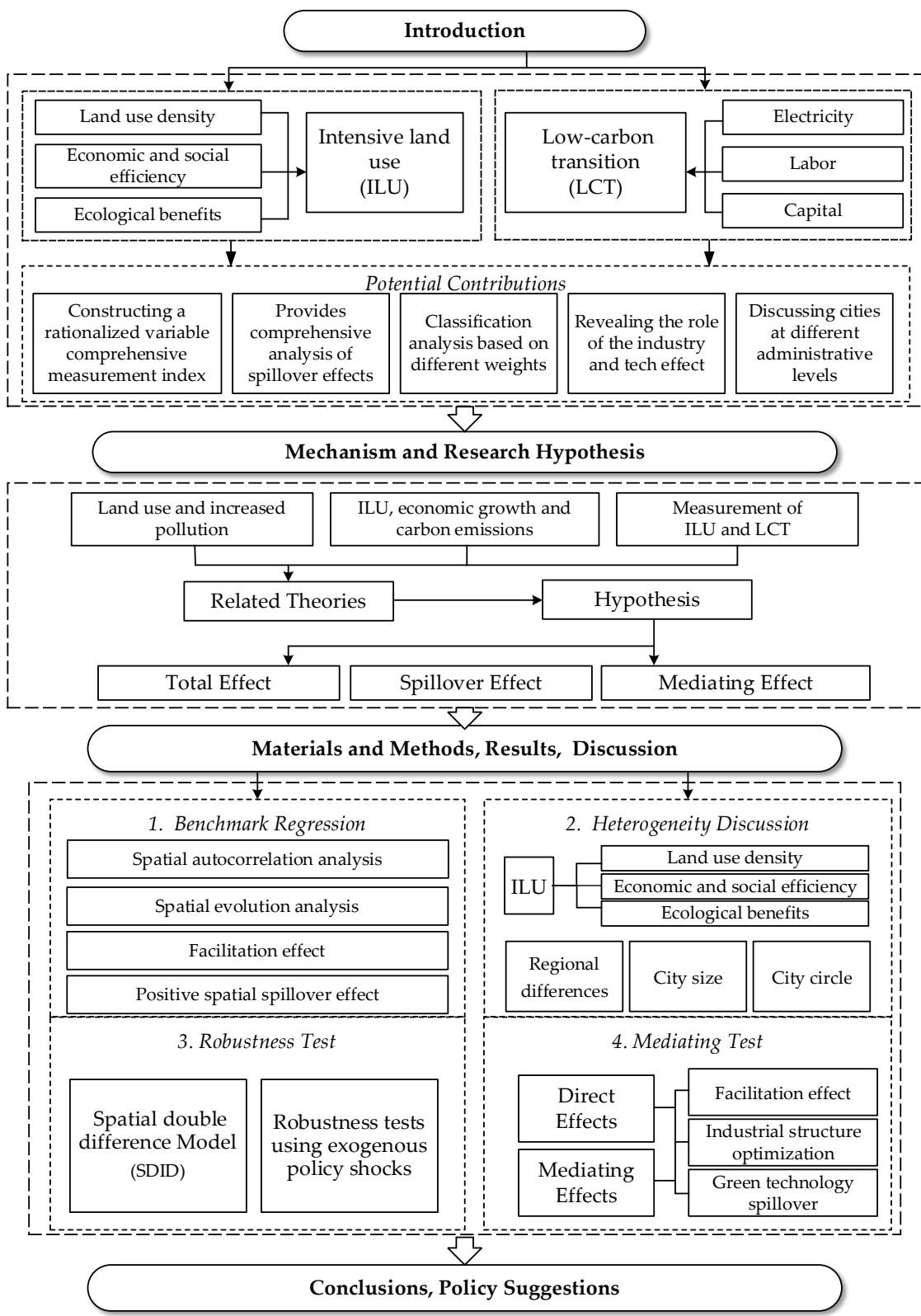

**Figure 2.** Outline of the research framework.

## 2. Mechanism and Research Hypothesis

*ILU* is a critical way to build an urban ecological civilization, focusing on conservation, efficiency, and ecology. Economic green transformation is the primary way to realize the goal of sustainable development, focusing on economic efficiency and carbon reduction [33]. Compared with traditional land use, *ILU* embodies the concept of scientific and clean development. Whether it is to reduce emission sources or increase carbon sink absorption, *ILU* plays an important role [34]. The mechanism between *ILU* and China's low-carbon transition can be analyzed in economic and ecological systems.

### 2.1. Promoting Effect of Intensive Land Use on Low-Carbon Transition

First, we consider the economic growth effect. *ILU* is to increase the input of factors such as capital, labor, and technology on the urban land stock and improve land use efficiency through rational layout and optimization of land use structure to promote sustainable development [35]. According to the law of increasing and decreasing land remuneration, before reaching the highest point of remuneration, the more capital and labor force invested in the land per unit area, the higher the economic output obtained, that is, the higher the intensity of land use and development, the higher the contribution to economic growth [36].

Second, we consider the carbon emission reduction effect. At the ecosystem level, *ILU* corresponds to the impact of land use change on soil carbon stock and vegetation carbon stock. According to the land use classification, construction land is the primary carbon source, while ecological and agricultural land are essential sources of carbon sinks. *ILU* effectively reduces the conversion of agricultural and ecological land, such as garden land, forest land, and grassland, to construction land and increases carbon sink absorption within the ecosystem [37]. In addition, at the economic system level, with the strengthening of land use constraints, low energy-consuming technologies, enterprises, and industrial chains will be "squeezed out", thus reducing carbon emissions. Compact land space pattern facilitates public transportation use and reduces infrastructure construction waste [38]. It helps to improve the efficiency of centralized energy supply and utilization and reduces the growth of carbon sources from land construction.

### 2.2. Spatial Spillover Mechanism

According to the theory of agglomeration effect, the increased density and spatial proximity of economic activities on land contribute to the economies of scale in production and transactions and resulting spillovers on a local scale [39]. Considering that carbon emissions are mainly influenced by socioeconomic drivers such as the stage of economic development, energy resource endowment, and consumption patterns. Land use mode, scale, structure, and intensity are closely related to industrial development status and technological progress [40]. Therefore, intensive land use affects the low-carbon transformation of neighboring cities mainly by influencing the socioeconomic constitutive factors reflected in industrial development and technological progress [41,42].

The mediating role of industrial structure transformation. On the one hand, as the land use pattern constrains the industrial layout, the crude land approach is challenging to promote the automatic transformation of industry. The theoretical connotation of *ILU* includes the principles of "structural optimization" and "market allocation." Conversely, *ILU* encourages the development of resource-saving and environment-friendly industries, which helps cities establish a modernized industrial system characterized by low energy consumption and emissions [40]. On the other hand, due to market competition and the price mechanism, industrial agglomeration forces enterprises to improve production technology and techniques and promotes upgrading the industrial chain layout [43]. As a result, with the gradual withdrawal of resource-intensive enterprises, the utility of energy utilization has been enhanced. The booming development of high-tech enterprises and service industries is conducive to easing resource dependence and environmental pressure. This transformation of industrial structure promotes the diversified division

of labor among enterprises, which helps the low-carbon production links and industrial clusters among cities to generate symbiosis, complement each other, and promote the low-carbon transformation and development of neighboring cities [44].

The intermediary role of technology spillover. The agglomeration economy formed by *ILU* makes cities accumulate innovative resources and also promotes technological overflow from cities. The theory of *ILU* includes the principles of "conservation first" and "reform and innovation," which help accelerate the R&D, innovation, and application of production, environmental protection, and energy-saving technologies and generate technological factor overflows from the region to neighboring or related regions. The overflow accelerates the learning and disseminating of green technologies and innovation among cities. Neighboring cities at this time can realize imitation and secondary innovation with the help of technological overflow, thus promoting green transformation.

Overall, spatial spillovers from economic systems and ecosystems diffuse the effects of NTUs on local industrial restructuring and technological spillovers to neighboring regions, which ultimately manifests itself in the diffusion of local *ILU*s' driving effects on neighboring regions' *LCT*. The above spillover mechanisms accumulate layer by layer, forming the total effect of *ILU* on the overall regional *LCT*.

**Hypothesis (H1).** *Intensive land use drives low-carbon transition and can promote neighboring cities to achieve low-carbon transition goals.*

**Hypothesis (H2).** *Intensive land use can promote low-carbon transition through industrial structure transformation and technology spillover.*

*2.3. Spatial Decay Mechanism*

Most studies recognize that spillover effects are characterized by spatial decay. It is because the cost of logistics and information exchange will rise with increased geographical distance and the restriction of administrative boundaries. As a result, the spatial spillover effect of *ILU* may show a specific attenuation pattern and boundary effect as the possibility of factor spillover decreases.

First, we consider the role of geographic distance. Studies have confirmed that information dissemination shows the law of attenuation with increased geographical distance. As the distance between neighboring cities and knowledge-center cities increases, the efficiency of information dissemination decreases [45]. Furthermore, increased geographic distance leads to interfirm transaction costs, transportation costs, and risk control. This former is a function of the geographic distance between the cooperating parties [46,47], and the latter is reflected in the increased risk of breach of contract and the reduced level of trust, among others. These are not conducive to inter-city technology, exchange, and cooperation, leading to spatial limitations in the spillover effects of *ILU* on low-carbon transitions in neighboring cities.

Second, we consider the role of administrative boundaries. Local governments in China can deeply participate in the urban economic growth model, which provides many administrative and financial resources needed to promote urban development. However, this model may cause local governments to ignore the big picture of policies and hinder the joint promotion of inter-city policies. Specifically, local governments may impose explicit or implicit administrative restrictions on factor mobility to protect local markets. Without an effective inter-regional coordination mechanism, it is not easy to realize the effective interconnection of cross-regional infrastructure. It will not be conducive to the optimal allocation of resources and the dissemination of advanced technologies, hindering interregional cooperation in production links and industrial synergistic development and causing the spatial spillover of *ILU* to form a specific border effect.

**Hypothesis (H3).** *Influenced by geographic distance and administrative boundaries, there is spatial attenuation in the spillover effect of intensive land use on low-carbon transition.*

## 3. Materials and Methods

### 3.1. Data Sources

This paper takes Chinese prefecture-level cities as the research object. After deleting the samples with missing severe data, we selected 283 prefecture-level cities in China (except Tibet, Hong Kong, Macao, and Taiwan) from 2006 to 2019 as the research sample. Variable data were obtained from *China Urban Statistical Yearbook* (2007–2020), *China Statistical Yearbook* (2007–2020), and statistics published by the National Bureau of Statistics. Green patent authorizations were obtained from the China Research Data Service Platform (CNRDS). In addition, we verified all data obtained to ensure accuracy, and some missing values were supplemented by interpolation. Some key variables were logarithmized to ensure consistent statistical caliber.

### 3.2. Research Method

#### 3.2.1. Spatial Autocorrelation Test

We refer to the study of Elhorst (2014) [48] for the first step of testing the applicability of the spatial measurement model to determine the spatial correlation of the variables. We choose *Moran's I* index, which is more common in the existing literature, to determine whether the spatial autocorrelation of low-carbon transition exists in each city. In addition, we also chose to use *Geary's C* index to conduct the spatial autocorrelation test from a global perspective. It is more sensitive than *Moran's I* index in the localized test (a value less than 1 indicates a positive correlation).

$$Moran's\ I = \sum_{i=1}^{n}\sum_{j=1}^{n} W_{ij}(X_i - \overline{X})(X_j - \overline{X}) / S^2 \sum_{i=1}^{n}\sum_{j=1}^{n} W_{ij} \tag{1}$$

$$Geary's\ C = (n-1)\sum_{i=1}^{n}\sum_{j=1}^{n} W_{ij}(X_i - X_j)^2 / 2\left(\sum_{i=1}^{n}\sum_{j=1}^{n} W_{ij}\right)\left[\sum_{i=1}^{n}(X_i - \overline{X})^2\right]$$
$$S^2 = \sum_{i=1}^{n}(X_i - \overline{X})^2 / n\ ; \quad \overline{X} = \frac{1}{n}\sum_{i=1}^{n} X_i \tag{2}$$

where $X_i$ and $X_j$ are the actual observed values of regions "*i*" and "*j*", respectively. $W_{ij}$ is the spatial weight matrix. *n* represents the total number of geography units that refers to the sample cities.

#### 3.2.2. Spatial Econometric Models

First, we construct neighborhood spatial weights ($W_{adj}$), inverse distance spatial weight matrix ($W_{dis}$), economic spatial weight matrix ($W_{econ}$), and economic distance spatial weight matrix ($W_{econdis}$) from two perspectives: geographic distance and economic distance, respectively. For the geographic distance spatial weight matrix, the closer the distance, the greater the influence of the neighborhood. For the economic distance spatial weight matrix, cities with higher economic levels significantly influence the neighborhood more than cities with lower economies.

$$w_{adj} = \begin{cases} 0 & (\text{urban area } i,\ j \text{ are not adjacent}) \\ 1 & (\text{urban area } i,\ j \text{ are adjacent}) \end{cases} \tag{3}$$

$$w_{dis} = \begin{cases} 0 & (i \neq j) \\ \frac{1}{d_{ij}} & (i = j), \end{cases} d_{ij} \text{ is the distance between two urban centers} \tag{4}$$

$$w_{econ} = \begin{cases} 0 & (i \neq j) \\ \frac{1}{|X_i - X_j|} & (i = j), \end{cases} X \text{ is the economic aggregate, measured using GDP} \tag{5}$$

$$w_{econdis} = aW_{dis} + bW_{econ}, \quad a = b = 0.5 \tag{6}$$

Spatial econometric model was used to identify and verify the spatial spillover effects of *ILU* on low-carbon transition. The spatial econometric model is constructed as follows:

$$LCT_{it} = \alpha + \rho \sum_{j=1,j\neq i}^{N} \omega_{ij} LCT_{jt} + \beta X_{it} + \theta \sum_{j=1}^{N} \omega_{ij} X_{ijt} + \varphi_i + \nu_t + \varepsilon_{it}$$

$$\varepsilon_{it} = \psi \sum_{j=1,j\neq i}^{N} \omega_{ij} \varepsilon_{jt} + \mu_{it}; \mu_{it} \sim N(0,\sigma^2 I) \tag{7}$$

where *LCT* denotes the urban low-carbon transition variable; *X* is each explanatory variable, including *ILU*; *w* is the spatial weight matrix; $\rho$ is the coefficient of the effect of the local explanatory variable on the explanatory variables in other urban areas. When $\rho > 0$, it indicates the spatial spillover effect between adjacent regions; when $\rho < 0$, it indicates the spatial negative effect between adjacent regions. $\beta$, $\theta$ are the parameter estimates of *X* explanatory variables; $\varphi$ and $\nu$ denote the area effect and time effect, respectively; $\psi$ denotes the spatial correlation between residuals; $\varepsilon$ is the random error term; *i*, *t* are the area individual and time dimensions.

In Equation (7), if $\rho = 0$, $\theta = 0$, $\psi = 0$, then Equation (7) is the spatial lag model (SLM), if $\rho = 0$, $\theta = 0$, $\psi \neq 0$, then it is the spatial error model (SEM), if $\rho \neq 0$, $\theta \neq 0$, $\psi = 0$, then it is the spatial Durbin model (SDM). In this paper, the Wald test and LR test are used to screen SLM, SEM, and SDM. If the tests both reject the original hypothesis of setting $H_0: \theta = 0$ and $H_0: \theta + \rho\beta = 0$, SDM is selected, and one of the original hypotheses is accepted, then the choice is made between SLM and SEM.

When the spatial Durbin model (SDM) was selected for the model, the direct, indirect, and total effects proposed by LeSage & Pace (2010) were used to further examine the spatial effects of the impact of *ILU* on urban low-carbon transition [49]. Using $Y_t$ to represent the *LCT* vector, the SDM model is rewritten into the following vector form.

$$Y_t = (1 - \rho W)^{-1}(\beta X_t + \theta W X_t) + (1 - \rho W)^{-1}\mu_t \tag{8}$$

In Equation (8), we derive the partial differential matrix by taking the *k* explanatory variable as the independent variable. The mean of the diagonal elements represents the average effect of the change in the explanatory variables on the explanatory variables of local areas, which is the direct effect. The mean of the non-diagonal elements represents the average effect of the change in the explanatory variables on the variables of the explanatory variables in other urban areas, which is the indirect effect.

$$\left[\frac{\partial Y}{\partial X_{1k}} \cdots \frac{\partial Y}{\partial X_{Nk}}\right]_t = (1 - \rho W)^{-1} \begin{bmatrix} \beta_k & \omega_{12}\nu_k & \cdots & \omega_{1N}\nu_k \\ \omega_{21}\nu_k & \beta_k & \cdots & \omega_{2N}\nu_k \\ \vdots & \vdots & \ddots & \vdots \\ \omega_{N1}\nu_k & \omega_{N2}\nu_k & \cdots & \beta_k \end{bmatrix} \tag{9}$$

### 3.2.3. Spatial Difference-in-Differences Model

According to the guideline of the Ministry of Land and Resources of China, the economic and intensive use of land is a strategic choice for new urbanization. Land-use intensification is an important policy tool in promoting the implementation of new urbanization. Therefore, this paper considers the new urbanization contention policy a proxy variable for *ILU*. In this study, we combine the difference-in-differences model (DID) with the spatial econometric model and relax the original assumption that the experimental group will not affect individuals in the control group. As a result, we construct the spatial difference-in-differences model (SDID) as follows:

$$CEE_{it} = \gamma_0 + \rho_1 \sum_{j=1,j\neq i}^{N} \omega_{ij} CEE_{jt} + \gamma_1 policy_{it} + \theta_1 \sum_{j=1}^{N} \omega_{ij} policy_{ijt} + \gamma_2 Control_{it} + \theta_2 \sum_{j=1}^{N} \omega_{ij} Control_{ijt} + \varphi_i + \nu_t + \varepsilon_{it} \tag{10}$$

where *policy* denotes the exogenous policy shock of "new urbanization"; $\gamma_1$ denotes the impact coefficient of policy on local low-carbon economic transition; $\theta_1$ denotes the estimated coefficient of policy on low-carbon economic transition in neighboring areas; the rest of variables are explained in the same way as Equation (7).

*3.3. Research Method*

3.3.1. Explained Variable

Low-carbon transition (*LCT*) is the target explanatory variable. Considering the low-carbon transition (*LCT*) should balance carbon dioxide emission reduction and economic development [50], which can obtain the maximum economic output with the least factor input and the lowest carbon emission. Therefore, this paper uses the efficiency tool to represent the urban low-carbon transition. Stochastic frontier analysis utilizing the "input-output" paradigm can effectively deal with efficiency issues. In this study, specific indicators are selected with reference to existing studies [51]. Among them, input indicators include energy, labor, and capital, using the urban electricity consumption to measure the electricity input, the total urban employment to measure the labor input [52], and the urban capital stock based on the perpetual inventory method to measure the capital input [53]. Output indicators include desired output and non-desired output, and the urban GDP is used to measure the desired output of the city. The indicator system is shown in Table 1.

**Table 1.** Indicator system of low-carbon transition.

| Indicator | Variable | Description |
|---|---|---|
| Input | Electricity | Urban total social electricity consumption (Unit: 10,000 kw-h) |
| | Labor | Total employment in the city (Unit: 10,000 persons) |
| | Capital | Urban capital deposit (Unit: 10,000 yuan) |
| Desired output | Economic efficiency output | Urban GDP (Unit: 10,000 yuan) |
| Non-desired output | Carbon dioxide emissions | Urban $CO_2$ emissions (Unit: 10,000 tons) |

We also refer to the IPCC 2006 methodology to measure the undesired urban output using the carbon emissions generated during the consumption of natural gas, liquefied petroleum gas, electricity, and thermal energy for the whole city community [54]. Combined with the methodology of Tone and Tsutsui (2010) [55], we measure *CCE* using an excess efficiency model (EBM).

$$r^* = \min\left(\theta - \varepsilon^- \sum_{i=1}^{m} \frac{\omega_i^- s_i^-}{x_{i0}}\right) \Big/ \left[\varphi + \varepsilon^+\left(\sum_{r=1}^{s} \frac{\omega_r^+ s_r^+}{y_{r0}} + \sum_{p=1}^{q} \frac{\omega_p^{u-} s_p^{u-}}{u_{p0}}\right)\right] \quad (11)$$

$$s.t. \begin{cases} \sum_{j=1}^{n} x_{ij}\lambda_j + s_i^- = \theta x_{i0} & (i = 1, 2 \ldots, m) \\ \sum_{j=1}^{n} y_{rj}\lambda_j - s_r^+ = \theta y_{r0} & (r = 1, 2 \ldots, s) \\ \sum_{j=1}^{n} u_{pj}\lambda_j + s_p^- = \theta u_{p0} & (p = 1, 2 \ldots, q) \\ \lambda_j \geq 0; s_i^-, s_r^+, s_p^- \geq 0 \end{cases} \quad (12)$$

In Equation (11), $r^*$ represents the optimal efficiency value of the *LCT* measured by the super total factor productivity model. There are $m + 1$ parameters in this model. $x$, $y$ and $u$ represent the inputs, expected outputs, and unexpected outputs of DMU0, respectively, $\theta$ is the radial efficiency value; $s$ represents the input slack vector. $\varepsilon$ is a core parameter that determines the importance of the non-radial part of the computation of the efficiency value of $r^*$, and it takes the value in the range of [0, 1]. When $\varepsilon = 0$, it is equivalent to the radial model, and when taking the value of 1, it is equivalent to the SBM model. In Equation (12), $\lambda$ denotes the weight coefficient.

### 3.3.2. Core Explanatory Variables

According to the different construction bases and standards of the evaluation index system, the evaluation index system mainly includes the "input-output", the "economic-social-ecological", and the "intensive-efficient-coordinated" index system. In selecting indicators, many scholars believe that the number of indicators is lower than possible but should be reasonably screened. Evaluation methods mainly include multiple single-indicator methods, factor synthesis evaluation methods, fuzzy synthesis evaluation methods, ideal value correction models, etc. Considering that intensive land use is a dynamic process, the degree of intensification can be effectively enhanced by improving the land use intensity and optimizing the land output efficiency and land use structure within a certain period. Based on the principle of dominant factors and local conditions, this paper selects indicators according to the evaluation system of "economy-society-ecology." Specific indicators and their measurement methods are shown in Table 2.

**Table 2.** Indicator system of intensive land use.

| Primary Indicators | Description (Unit) | Indicator Attributes |
|---|---|---|
| Land use density | Built-up area/total urban area (%) | Negative |
| | Capital stock/built-up area (yuan/KM$^2$) | Positive |
| | Road area/built-up area (m$^2$/KM$^2$) | Positive |
| | House area/built-up area (m$^2$/KM$^2$) | Positive |
| | Urban population/built-up area (10,000 people/KM$^2$) | Positive |
| | Employment/built-up area (person/KM$^2$) | Positive |
| Economic and social efficiency | GDP/built-up area (yuan/KM$^2$) | Positive |
| | Non-agricultural industry output value/built-up area (yuan/KM$^2$) | Positive |
| | General income of fiscal budget/built-up area (yuan/KM$^2$) | Positive |
| | Disposable income of urban residents/built-up area (yuan/KM$^2$) | Positive |
| | Built-up area/urban resident population (m$^2$/person) | Negative |
| Ecological benefits | Per capita green area (m$^2$/person) | Positive |
| | Green coverage rate of urban built-up areas (%) | Positive |
| | Industrial sewage discharge per capita (ton/person) | Negative |

In this paper, the TOPSIS model is chosen to measure the *ILU* composite indicators. TOPSIS model is an effective method in multi-objective decision analysis. It is a ranking method close to the ideal solution, which ranks the indicators by detecting the distance between the evaluation object and the optimal solution and the worst solution. In the calculation process, it is necessary to normalize the positive and negative indicators separately.

$$C_i = \sqrt{\sum_{j=1}^{m} (Z_{\text{min}j} - Z_{ij})^2} / \left( \sqrt{\sum_{j=1}^{m} (Z_{\text{max}j} - Z_{ij})^2} + \sqrt{\sum_{j=1}^{m} (Z_{\text{min}j} - Z_{ij})^2} \right) \quad (13)$$

$$Z_{ij} = X_{ij} / \sqrt{\sum_{i=1}^{n} X_{ij}^2} \quad or \quad Z_{ij} = \frac{1}{X_{ij}} / \sqrt{\sum_{i=1}^{n} \left( \frac{1}{X_{ij}} \right)^2}$$

$$Z^+ = (Z_{\text{max}1} Z_{\text{max}2} Z_{\text{max}3} \ldots Z_{\text{max}m})$$

$$Z^- = (Z_{\text{min}1} Z_{\text{min}2} Z_{\text{min}3} \ldots Z_{\text{min}m}) \quad (14)$$

where *n* is the number of cities participating in the evaluation; *m* is the number of evaluation indicators. $C_i$ denotes the proximity of evaluation object *i* to the optimal solution, and finally, the comprehensive evaluation results of *ILU* in Chinese cities are obtained by sorting them according to the size of $C_i$.

### 3.3.3. Control Variables

In this study, other factors that may affect the low-carbon urban transition are included in the empirical model in order to mitigate omitted variable bias as much as possible. The main ones include. For environmental regulation (*ER*), we choose three indicators: sulfur dioxide removal rate (industrial sulfur dioxide removal/industrial sulfur dioxide generation), industrial soot removal rate (industrial soot removal/industrial soot generation), and comprehensive industrial solid waste utilization rate (comprehensive industrial solid waste utilization/(comprehensive industrial solid waste generation + comprehensive utilization of previous years' storage)), and use the entropy value method to calculate the intensity of environmental regulation. For the annual average temperature (*TEM*), we used the cumulative daily temperature average to represent this variable. Openness to foreign investment (*OPEN*), we use the annual real foreign investment (converted to RMB based on the average RMB exchange rate) as a share of GDP. Government intervention (*GOV*), we use the share of fiscal expenditure net of science and education in total fiscal expenditure. In industrial agglomeration (*AGG*), we use the Location Quotient method to calculate the manufacturing agglomeration status of each city. Marketization (MARK), which we measure using the share of self-employment and private employment in total employment. Financial development (*FIAN*), which we measure using the year-end loan balance as a share of GDP.

### 3.3.4. Other Variables

1. Instrumental variable

In this study, we will further examine the implementation effect of the new urbanization pilot policy (*Policy*) in the robustness test. In 2013, China established a new "people-oriented" urbanization policy. In 2015 and 2016, China's National Development and Reform Commission (NDRC) announced three batches of comprehensive national pilot projects for new urbanization. Since then, Chinese government departments have continued to improve the program and expand the pilot project scope into developing replicable and replicable experiences. This study assigns a value of 1 to the approved pilot cities (experimental group) and 0 to the unapproved non-pilot cities (control group), denoted as $Treated_i$. Among the pilot cities, this study assigns a value of 1 to the year in which the pilot cities are approved and subsequent years and 0 to the remaining years. All year's corresponding to the non-pilot cities are assigned a value of 0 and denoted as $Time_t$. In this case, $Policy_{it} = Treated_i * Time_t$.

2. Channel variables

Based on the theoretical analysis in the previous subsections, the two key mechanism variables for channel analysis are industrial structure transformation (*IS*) and technology spillover (*TS*).

Industrial structure transformation (*IS*). The upgrading of industrial structures towards cleanliness is the key to realizing the goal of green development. The current indicators for industrial structure upgrading mainly use internal structure change, energy consumption per unit GDP of industry, and product sales of pollution-intensive industries. We utilize the entropy value method to determine the degree of cleaner transformation of industrial structure. In this paper, we refer to Zhang et al. (2023) to construct the indicator system from two aspects of clean energy consumption and clean production [31]. Clean energy consumption is measured by the ratio of total industrial energy consumption to industrial added value; clean production is expressed by the ratio of regional industrial added value to carbon emissions. Through the dimensionless quantization of the indicators, the entropy value method is then used to identify the degree of cleaner transformation of the industrial structure.

Technology spillover (*TS*). Generally speaking, due to China's imperfect patent guarantee mechanism and relatively backward R&D capability, it is difficult for enterprises to convert R&D inputs into green innovation outputs. In contrast, the number of green

patents can reflect the actual innovation outputs more objectively. We use the number of green patent acquisitions obtained in one year as a proxy variable for technology spillovers.

The results of descriptive statistics for each variable are shown in Table 3.

**Table 3.** Descriptive statistics of variables.

| Variable | Mean | N | SD | Min | p25 | p50 | p75 | Max |
|----------|------|---|-----|-----|-----|-----|-----|-----|
| LCT | 0.500 | 3962 | 0.150 | 0.130 | 0.400 | 0.480 | 0.580 | 1.160 |
| ILU | 0.0800 | 3962 | 0.0700 | 0.0100 | 0.0500 | 0.0600 | 0.100 | 0.630 |
| ER | 0.610 | 3962 | 0.200 | 0.0600 | 0.460 | 0.660 | 0.760 | 0.990 |
| TEM | 14.60 | 3962 | 5.100 | −1.090 | 10.91 | 15.54 | 17.90 | 25.68 |
| OPEN | 1.900 | 3962 | 1.980 | 0 | 0.460 | 1.280 | 2.690 | 15.32 |
| GOV | 0.800 | 3962 | 0.0400 | 0.610 | 0.780 | 0.800 | 0.830 | 0.980 |
| AGG | 0.860 | 3962 | 0.480 | 0.0200 | 0.520 | 0.770 | 1.140 | 3.050 |
| MAK | 0.480 | 3962 | 0.140 | 0 | 0.380 | 0.480 | 0.580 | 0.940 |
| FIAN | 0.880 | 3962 | 0.560 | 0.0800 | 0.540 | 0.710 | 1.010 | 9.620 |

## 4. Results

### 4.1. Baseline Regression Analysis

#### 4.1.1. Spatial Autocorrelation Test and Spatio-Temporal Distribution

We calculated the spatial correlation of *LTC* in China using ArcGis 10.2 software. As shown in Appendix A Table A2, the spatial correlation indices were significant for all years. Among them, the global *Moran's I* value are all greater than zero, and the *Geary's C* values are all within the interval [0, 1]. It indicates a significant positive spatial correlation of *LCT* at the four-city level in China. In the time dimension, the global *Moran's I* index increases yearly, revealing that the spatial correlation of low carbon transition among cities has been strengthened year by year in recent years.

#### 4.1.2. Baseline Result

We first performed the Wald and LR tests, and the results showed that both passed the 1% significance test. This result rejects the original hypothesis of using the SLM or SEM model, indicating that the spatial error and lag terms exist simultaneously. Therefore, we use the spatial Durbin model for the empirical analysis. The Hausman test results pass the 1% significance test, indicating that the selection of the fixed-effects model is consistent with the model set. Table 4 reports the regression results of the spatial Durbin model for the four spatial weight matrices (regression results of Equation (7)). First, the coefficient of the effect of *ILU* on urban low-carbon transition is significantly positive under all four weights. It indicates a significant positive relationship between *ILU* and urban low-carbon efficiency.

Second, the results from the decomposition coefficients are shown. The results in column (2) of Table 4 show that the estimated coefficient of the indirect effect of *ILU* is 1.995 under the neighboring weights, which is significantly positive at the 1% statistical level. It indicates that *ILU* contributes to the local low-carbon transition and has a significant positive spatial spillover effect on the low-carbon transition of cities adjacent to the local one. The results from column (5) show that the indirect effect of *ILU* under geographical weight is 8.195 and passes the significance level test at the 1% level. The results in columns (8) and (11) show that the indirect effect of *ILU* under the economic weight is 3.710, and the indirect effect of *ILU* under the economic distance weight is 8.224. Both coefficients are significant at the 1% level. Specifically, *ILU* has a significant positive spillover effect on urban low-carbon transition regardless of the spatial weights used.

Finally, in terms of coefficient magnitude, the indirect effect of *ILU* on urban low-carbon transition shows a consistent feature across all four spatial weights, i.e., the indirect effect coefficient is higher than the direct effect coefficient. It indicates that we need to test the relationship between *ILU* and urban low-carbon transition based on the perspective of spatial spillover. This facilitative spillover effect has a stronger explanatory power in the total effect.

**Table 4.** Results of spatial spillover effects.

| Panel A: Results of Adjacent Space Weights and Distance Spatial Weights. | | | | | |
|---|---|---|---|---|---|
| | $W_{adj}$ (Model 1) | | | $W_{dis}$ (Model 2) | | |
| **Variable** | **(1)** | **(2)** | **(3)** | **(4)** | **(5)** | **(6)** |
| | **LR_Direct** | **LR_Indirect** | **LR_Total** | **LR_Direct** | **LR_Indirect** | **LR_Total** |
| *ILU* | 0.6155 *** | 1.9447 *** | 2.5602 *** | 0.5692 *** | 8.1948 *** | 8.7640 *** |
| | (−4.54) | (−8.983) | (−11.114) | (−4.192) | (−4.726) | (−5.102) |
| *ER* | −0.0363 ** | 0.0779 ** | 0.0416 | −0.0395 ** | 0.7691 *** | 0.7296 *** |
| | (−2.226) | (−2.439) | (−1.215) | (−2.438) | (−2.833) | (−2.683) |
| *TEM* | −0.0108 | −0.0155 | −0.0263 *** | −0.0086 | −0.0402 | −0.0489 |
| | (−0.971) | (−1.311) | (−4.250) | (−0.923) | (−1.110) | (−1.627) |
| *OPEN* | −0.0032 * | −0.0016 | −0.0048* | −0.0043 ** | 0.0317 * | 0.0274 * |
| | (−1.726) | (−0.526) | (−1.753) | (−2.449) | (−1.932) | (−1.718) |
| *GOV* | −0.4000 *** | 0.3100 ** | −0.09 | −0.3665 *** | −0.011 | −0.3775 |
| | (−4.669) | (−2.311) | (−0.668) | (−4.290) | (−0.013) | (−0.461) |
| *AGG* | −0.0260 *** | 0.0162 | −0.0098 | −0.0326 *** | 0.1029 | 0.0703 |
| | (−3.490) | (−1.215) | (−0.741) | (−4.406) | (−1.28) | (−0.887) |
| *MAK* | 0.0301 | 0.0093 | 0.0394 | 0.0159 | 0.3887 | 0.4046 |
| | (−1.206) | (−0.201) | (−0.79) | (−0.64) | (−1.061) | (−1.102) |
| *FIAN* | −0.0383 *** | −0.0549 *** | −0.0932 *** | −0.0324 *** | −0.4038 *** | −0.4362 *** |
| | (−5.824) | (−4.227) | (−7.246) | (−4.928) | (−4.248) | (−4.628) |
| City FE | YES | YES | YES | YES | YES | YES |
| Year FE | YES | YES | YES | YES | YES | YES |
| Observations | 3962 | 3962 | 3962 | 3962 | 3962 | 3962 |
| R-squared | 0.542 | 0.572 | 0.572 | 0.445 | 0.673 | 0.491 |

| Panel B: Results of Economic Spatial Weights and Economic Distance Spatial Weights. | | | | | |
|---|---|---|---|---|---|
| | $W_{econ}$ (Model 3) | | | $W_{econdis}$ (Model 4) | | |
| **Variable** | **(7)** | **(8)** | **(9)** | **(10)** | **(11)** | **(12)** |
| | **LR_Direct** | **LR_Indirect** | **LR_Total** | **LR_Direct** | **LR_Indirect** | **LR_Total** |
| *ILU* | 0.4631 *** | 3.7100 *** | 4.1732 *** | 0.5688 *** | 8.2240 *** | 8.7928 *** |
| | (3.532) | (8.331) | (8.967) | (4.189) | (4.730) | (5.105) |
| *ER* | −0.0452 *** | 0.1454 ** | 0.1002 | −0.0395 ** | 0.7717 *** | 0.7323 *** |
| | (−2.791) | (2.473) | (1.562) | (−2.435) | (2.835) | (2.686) |
| *TEM* | −0.0246 *** | 0.0446 ** | 0.0200 | −0.0087 | −0.0399 | −0.0485 |
| | (−4.495) | (2.326) | (0.946) | (−0.931) | (−1.096) | (−1.612) |
| *OPEN* | −0.0025 | −0.0189 *** | −0.0214 *** | −0.0043 ** | 0.0320 * | 0.0277 * |
| | (−1.538) | (−3.008) | (−3.189) | (−2.445) | (1.942) | (1.728) |
| *GOV* | −0.2572 *** | −0.9358 *** | −1.1930 *** | −0.3656 *** | −0.0158 | −0.3814 |
| | (−3.249) | (−3.884) | (−4.579) | (−4.280) | (−0.019) | (−0.465) |
| *AGG* | −0.0230 *** | 0.0713 *** | 0.0483 * | −0.0327 *** | 0.1036 | 0.0709 |
| | (−3.244) | (2.692) | (1.750) | (−4.411) | (1.285) | (0.893) |
| *MAK* | 0.0255 | −0.0119 | 0.0136 | 0.0159 | 0.3878 | 0.4037 |
| | (1.031) | (−0.150) | (0.155) | (0.639) | (1.057) | (1.098) |
| *FIAN* | −0.0493 *** | 0.0067 | −0.0426 ** | −0.0324 *** | −0.4051 *** | −0.4376 *** |
| | (−7.795) | (0.341) | (−2.043) | (−4.928) | (−4.247) | (−4.625) |
| City FE | YES | YES | YES | YES | YES | YES |
| Year FE | YES | YES | YES | YES | YES | YES |
| Observations | 3962 | 3962 | 3962 | 3962 | 3962 | 3962 |
| R-squared | 0.470 | 0.309 | 0.498 | 0.527 | 0.559 | 0.558 |

Note: The numbers in parentheses are robust t-statistics. ***, ** and * represent significance levels of 1%, 5% and 10%, respectively.

Regarding the effects of other control variables: Environmental regulation (*ER*) shows consistent characteristics across the four spatial weights. Its direct effect coefficient (effect on local low-carbon transition) is significantly negative yet. However, the indirect effect coefficient (effect on the low-carbon transition of neighboring and associated cities) is significantly positive. The possible reason for this is that environmental regulation policies

emphasize the role of regulation, which reduces the total local pollution index while also impacting the development of local industries, thus showing a negative effect on low carbon efficiency. Further, under a more stringent environmental regulation policy, polluting industries in cities are at risk of closing down. It can force the development of local clean technologies. With the spillover of knowledge and technology, neighboring or associated cities can promote low-carbon transition development through secondary innovation and technology imitation. The indirect effect of government intervention (*GOV*) is significantly positive (0.310) with the neighboring weights and significantly negative (0.936) with the economic weights. The possible reason is that geographical proximity increases the "demonstration effect" of urban areas. The prosperity of local economic development can lead to technological progress and knowledge accumulation in neighboring cities, thus promoting the low-carbon transition of neighboring cities. Regarding economic distance, the development of local cities tends to eliminate pollution-intensive industries and make them move to cities with higher economic connectivity. It causes an increase in carbon emissions in the receiving region, which harms its green transformation. Financial development (*FIAN*) shows a significant indirect effect under the neighborhood, geographic, and economic geography weight. Moreover, it has a negative sign of the coefficient. Further deepening of financial development may have a siphoning effect on the neighboring areas' financial resources and human capital. This effect is not conducive to the expansion of production of the plot industries in the neighboring and associated cities nor to the updating and R&D of clean technologies. It will eventually lead to the inhibition of their low-carbon transition.

*4.2. Robustness Tests*

We first conduct a parallel trend test to analyze whether policy evaluation can be conducted using the double difference approach. After the results showed that this important test was passed, we proceeded to model estimation. Table 5 reports the regression results of the spatial difference-in-differences model for the four spatial weight matrices (regression results of Equation (10)). As we can see from Table 5, the indirect effects of the *ILU* ("New Urbanization" pilot policy dummy variables) are significant at different spatial weight matrices, and the coefficients of the variables are positive. It implies that the *ILU* construction will significantly promote the low-carbon transformation of the surrounding and associated cities. It also indicates a non-negligible spatial correlation in the error term of the model, and if not taken into account, the regression results will produce biased estimates. Using spatial econometric models in this paper is necessary.

In particular, there are some characteristics of Models 1–4 results based on spatial weights of different geographical elements. As can be seen, the coefficient of distance spatial weights ($W_{dis}$, 0.342) and the coefficient of e economic distance spatial weights ($W_{econdis}$, 0.343) are higher than the coefficient of an indirect effect of economic, spatial weights ($W_{econ}$, 0.112) and higher than the coefficient of neighboring spatial weights ($W_{adj}$, 0.051). It may be because the pilot cities have made more efforts to promote low-carbon, green, inclusive, and intelligent cities. The demonstration effect and the economic correlation effect on the neighboring cities and associated cities exemplify the positive effect on the low-carbon transition of the cities. Furthermore, the size of the coefficient indicates that this role-modeling effect is more likely to be constrained by geographical distance.

*4.3. Mechanism Verification*

4.3.1. Channel Mechanism of Industrial Structure Transformation

Model 1 in Table 6 shows the impact of intensive land use on industrial structure upgrading, and it can be seen that the impact is positive and significant, indicating that *ILU* promotes China's low-carbon transformation through the *IS* influence mechanism. Specifically, *ILU*, as a long-term national policy, will constrain the disorderly expansion of enterprise land use and promote the economization of enterprise production and operation in the coming period. In addition, the objective constraints of *ILU* on urban space will

limit the entry of highly polluting and low-value-added industries. Therefore, in cities with intensive land utilization, industries can obtain green development and regulate the industrial layout by regulating the proportion of clean industries, thus promoting low-carbon development.

**Table 5.** Robustness test results of exogenous shock.

| Panel A: Results of Adjacent Space Weights and Distance Spatial Weights. | | | | | | |
|---|---|---|---|---|---|---|
| **Variable** | $W_{adj}$ **(Model 1)** | | | $W_{dis}$ **(Model 2)** | | |
| | **(1)** | **(2)** | **(3)** | **(4)** | **(5)** | **(6)** |
| | **LR_Direct** | **LR_Indirect** | **LR_Total** | **LR_Direct** | **LR_Indirect** | **LR_Total** |
| ILU | 0.0051 | 0.0510 *** | 0.0561 *** | −0.0073 | 0.3422 *** | 0.3349 *** |
| | (0.507) | (3.500) | (4.082) | (−0.731) | (3.702) | (3.723) |
| ER | −0.0500 *** | 0.0425 | −0.0075 | −0.0427 *** | 0.6401 ** | 0.5975 * |
| | (−3.046) | (1.263) | (−0.206) | (−2.614) | (2.062) | (1.919) |
| TEM | −0.0117 | −0.0120 | −0.0237 *** | −0.0098 | −0.0382 | −0.0481 |
| | (−1.048) | (−0.999) | (−3.663) | (−1.048) | (−0.938) | (−1.389) |
| OPEN | −0.0030 | −0.0083 *** | −0.0113 *** | −0.0053 *** | −0.0002 | −0.0055 |
| | (−1.628) | (−2.637) | (−4.000) | (−2.975) | (−0.013) | (−0.321) |
| GOV | −0.4750 *** | 0.0111 | −0.4639 *** | −0.4279 *** | −0.7720 | −1.1999 |
| | (−5.595) | (0.083) | (−3.450) | (−5.117) | (−0.828) | (−1.310) |
| AGG | −0.0255 *** | 0.0113 | −0.0141 | −0.0309 *** | 0.1323 | 0.1014 |
| | (−3.389) | (0.820) | (−1.015) | (−4.162) | (1.416) | (1.095) |
| MAK | 0.0294 | −0.0105 | 0.0190 | 0.0100 | 0.0321 | 0.0420 |
| | (1.165) | (−0.218) | (0.362) | (0.399) | (0.076) | (0.099) |
| FIAN | −0.0388 *** | −0.0567 *** | −0.0955 *** | −0.0311 *** | −0.4983 *** | −0.5294 *** |
| | (−5.884) | (−4.224) | (−7.054) | (−4.715) | (−4.233) | (−4.516) |
| City FE | YES | YES | YES | YES | YES | YES |
| Year FE | YES | YES | YES | YES | YES | YES |
| Observations | 3962 | 3962 | 3962 | 3962 | 3962 | 3962 |
| R-squared | 0.475 | 0.440 | 0.433 | 0.587 | 0.502 | 0.588 |
| **Panel B: Results of Economic Spatial Weights and Economic Distance Spatial Weights.** | | | | | | |
| **Variable** | $W_{econ}$ **(Model 3)** | | | $W_{econdis}$ **(Model 4)** | | |
| | **(7)** | **(8)** | **(9)** | **(10)** | **(11)** | **(12)** |
| | **LR_Direct** | **LR_Indirect** | **LR_Total** | **LR_Direct** | **LR_Indirect** | **LR_Total** |
| ILU | 0.0179 ** | 0.1121 *** | 0.1300 *** | −0.0073 | 0.3438 *** | 0.3365 *** |
| | (2.008) | (3.555) | (3.826) | (−0.735) | (3.707) | (3.727) |
| ER | −0.0534 *** | 0.1415 ** | 0.0881 | −0.0426 *** | 0.6422 ** | 0.5996 * |
| | (−3.257) | (2.267) | (1.292) | (−2.613) | (2.063) | (1.920) |
| TEM | −0.0207 *** | 0.0664 *** | 0.0457 ** | −0.0099 | −0.0377 | −0.0476 |
| | (−3.747) | (3.300) | (2.059) | (−1.058) | (−0.922) | (−1.371) |
| OPEN | −0.0045 *** | −0.0317 *** | −0.0362 *** | −0.0052 *** | −0.0001 | −0.0054 |
| | (−2.804) | (−4.866) | (−5.227) | (−2.970) | (−0.006) | (−0.313) |
| GOV | −0.3906 *** | −1.5333 *** | −1.9239 *** | −0.4269 *** | −0.7834 | −1.2102 |
| | (−5.038) | (−6.378) | (−7.540) | (−5.105) | (−0.838) | (−1.319) |
| AGG | −0.0226 *** | 0.0578 ** | 0.0352 | −0.0310 *** | 0.1334 | 0.1024 |
| | (−3.145) | (2.068) | (1.203) | (−4.167) | (1.423) | (1.103) |
| MAK | 0.0101 | −0.0701 | −0.0601 | 0.0099 | 0.0286 | 0.0386 |
| | (0.404) | (−0.839) | (−0.654) | (0.397) | (0.067) | (0.090) |
| FIAN | −0.0458 *** | 0.0243 | −0.0215 | −0.0311 *** | −0.4998 *** | −0.5309 *** |
| | (−7.197) | (1.168) | (−0.974) | (−4.715) | (−4.227) | (−4.508) |
| City FE | YES | YES | YES | YES | YES | YES |
| Year FE | YES | YES | YES | YES | YES | YES |
| Observations | 3962 | 3962 | 3962 | 3962 | 3962 | 3962 |
| R-squared | 0.533 | 0.552 | 0.522 | 0.455 | 0.662 | 0.431 |

Note: The numbers in parentheses are robust t-statistics. ***, ** and * represent significance levels of 1%, 5% and 10%, respectively.

**Table 6.** Channel mechanism verification.

| Variable | IS | TS |
|---|---|---|
| | **Model 1** | **Model 2** |
| *ILU* (Main) | 0.022 ** | 0.071 *** |
| | (2.11) | (5.18) |
| *ILU·W* | 0.156 *** | 4.966 *** |
| | (4.94) | (2.99) |
| Control | YES | YES |
| City FE | YES | YES |
| Year FE | YES | YES |
| Observations | 3962 | 3962 |

Note: The numbers in parentheses are robust t-statistics. *** and ** represent significance levels of 1% and 5%, respectively.

### 4.3.2. Channel Mechanism of Technology Spillover

As shown in Model 2 in Table 6, the effect of intensive land use on technology spillovers is significant, indicating that *ILU* promotes China's low-carbon transition through the influence mechanism of *TS*. Specifically, *ILU*'s land use restrictions on enterprises can first force enterprises to increase investment and research in green products and new materials. Second, compared with the standardized and large-scale production of the secondary industry, the knowledge, and technology-intensive tertiary sector tends to have higher value-added, lower energy consumption, and is more in line with the need for intensive land use. It means cities with intensive land utilization have built a good platform for technology R&D and dissemination. Third, the positive psychological effect of *ILU* on the low-carbon development of industries should not be ignored. Positive public opinion encourages regional industrial enterprises and regional enterprises to imitate each other and technological innovation, thus promoting the low-carbon transformation of the region.

### 4.4. Test of Spatial Attenuation Boundary

In order to examine the regional boundaries of *ILU* on urban low-carbon transition, this section uses threshold inverse distance spatial weights for multiple spatial Durbin model estimation analysis, and the weights are specifically set as follows:

$$w_{ij} = \begin{cases} 0 & (i = j \ or \ d_\mu > d_{ij} \ or \ d_{ij} > d_l \\ \frac{1}{d_{ij}} & (i \neq j \ and \ d_\mu < d_{ij} < d_l) \end{cases} \tag{15}$$

In Equation (15), $d_{ij}$ represents the distance between city $i$ and city $j$ regions, $d_\mu$ is the lower limit of spatial threshold distance, and $d_l$ is the upper limit of spatial threshold distance, whose values are set autonomously. When the distance between two cities is within the spatial threshold range, the spatial relationship between the two cities is considered to exist, and the weight value is the inverse of the distance between the two. Below or above this range, it is considered that there is no spatial relationship between the two cities, and its weight value is 0. Thus, we derive the spatial weight matrix of distances in different distance ranges and re-estimate the spatial Durbin model to measure the spatial spillover effects of *ILU* affecting urban low-carbon transition in different distance ranges (as shown in Table 7 and Figure 3).

The relationship between the local effect, spillover effect, and the corresponding distance threshold of *ILU* on urban low-carbon transition is shown in Figure 3. The effect of *ILU* on local low-carbon transition is relatively stable. However, the spillover effect exhibits an inverted "U" shape concerning the increase of distance threshold. It indicates that although *ILU* can promote the low-carbon transformation of neighboring and associated cities across space with their demonstration effects, their spillover effects have geographical peaks and decay boundaries due to the limitations of industrial development, infrastructure coverage, and administrative division. Moreover, the coefficient of the spillover effect peaks

at 250 km and then decays, while the indirect effect at 450 km is not significant. It implies that the *ILU* has the highest facilitation effect at 250 km. This radiation range is close to the radius of the regional scope of Chinese provinces, indicating that strengthening the *ILU* cooperation among regional urban clusters will help promote regional low-carbon transition.

**Table 7.** Spatial attenuation coefficients.

| Distance (KM) | LR_Direct | LR_Indirect | LR_Total |
|---|---|---|---|
| 50 | 0.7916 *** | 0.1158 ** | 0.9073 *** |
|  | (6.033) | (2.239) | (6.583) |
| 100 | 0.6747 *** | 0.4863 *** | 1.1611 *** |
|  | (5.066) | (4.274) | (7.150) |
| 150 | 0.5934 *** | 1.3236 *** | 1.9170 *** |
|  | (4.439) | (7.235) | (9.276) |
| 200 | 0.6166 *** | 1.7716 *** | 2.3882 *** |
|  | (4.633) | (7.057) | (8.862) |
| 250 | 0.6151 *** | 2.0986 *** | 2.7137 *** |
|  | (4.622) | (7.318) | (8.965) |
| 300 | 0.6511 *** | 0.7324 *** | 1.3835 *** |
|  | (4.671) | (3.002) | (5.638) |
| 350 | 0.6601 *** | 0.7008 *** | 1.3609 *** |
|  | (4.740) | (2.784) | (5.417) |
| 400 | 0.6847 *** | 0.6299 ** | 1.3146 *** |
|  | (4.927) | (2.427) | (5.060) |
| 450 | 0.7149 *** | 0.5342 ** | 1.2491 *** |
|  | (5.155) | (1.970) | (4.588) |
| 500 | 0.7441 *** | 0.3775 | 1.1216 *** |
|  | (5.351) | (1.355) | (4.054) |
| 550 | 0.7544 *** | 0.3050 | 1.0594 *** |
|  | (5.426) | (1.059) | (3.704) |
| 600 | 0.7519 *** | 0.3038 | 1.0557 *** |
|  | (5.412) | (1.044) | (3.656) |
| 650 | 0.7449 *** | 0.3561 | 1.1010 *** |
|  | (5.370) | (1.178) | (3.658) |
| 700 | 0.7509 *** | 0.3165 | 1.0674 *** |
|  | (5.415) | (1.044) | (3.538) |
| 750 | 0.7632 *** | 0.2695 | 1.0327 *** |
|  | (5.510) | (0.888) | (3.423) |

Note: The numbers in parentheses are robust t-statistics. *** and ** represent significance levels of 1% and 5%, respectively.

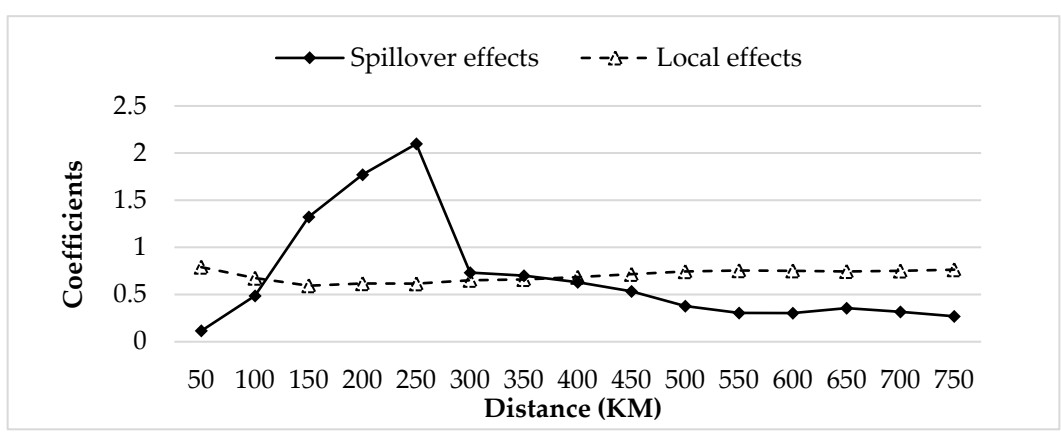

**Figure 3.** Regional boundary of spatial spillover effects.

*4.5. Heterogeneity Tests*

4.5.1. Heterogeneity at the *ILU* Dimension

In the previous section, we designed a comprehensive indicator to represent *ILU* based on the connotation of "intensive land use". The indicator contains three dimensions: density of land use, economic and social efficiency, and ecology benefits. In order to further analyze the influence mechanism of *ILU* on neighboring and associated towns, we conducted empirical tests on each of the three dimensions of *ILU*. The regression results are shown in Table 8.

**Table 8.** Heterogeneity test results of the *ILU* dimension.

| Variable | UCR | UCR | UCR |
|---|---|---|---|
| | (1) | (2) | (3) |
| | LR_Direct | LR_Indirect | LR_Total |
| density of land use | 0.3838 | 23.2338 *** | 23.6175 *** |
| | (1.096) | (4.245) | (4.299) |
| economic and social efficiency | 0.8405 *** | 20.7071 *** | 21.5477 *** |
| | (2.894) | (3.656) | (3.797) |
| ecology benefits | 4.1700 *** | 75.0155 *** | 73.8306 *** |
| | (10.986) | (3.301) | (3.208) |
| Control | YES | YES | YES |
| City FE | YES | YES | YES |
| Year FE | YES | YES | YES |
| Observations | 3962 | 3962 | 3962 |

Note: The numbers in parentheses are robust t-statistics. *** represent significance levels of 1%, respectively.

The density of land use provides a good impetus to the low-carbon transition of cities (direct effect of 0.384 and indirect effect of 23.234). However, the direct effect is not significant. This suggests that the density of land has a better demonstration effect on neighboring and associated cities, but attention needs to be paid to the local low-carbon transition. *ILU* in the land, social and ecological dimensions have significant positive spatial spillover. It shows that the rational and efficient use of land, increased social inclusion, and strengthened ecological constraints can form positive spillover to neighboring and associated cities. Among them, the coefficient of ecological dimension is the highest, at 75.016.

4.5.2. Spatial Heterogeneity

We further examine the spatial heterogeneity of the spillover effects of *ILU* on the low-carbon transformation of cities. We specifically focus on three perspectives: geographic location, city level, and city circle (economic location). Since this part of the test mainly considers the effect of spatial heterogeneity, we use geographic weights for estimation.

Cities with different geographic locations. As for the natural location of the province, different regions have distinct economic development goals, land use regulations, and contaminant pressure. Accordingly, the impact of IER may be affected by the geographical location, so we have divided the sample into three sample subgroups, including eastern, western, and central regions. The results are shown in Panel A of Table 9. In the eastern region, the spillover effect of *ILU* on urban low-carbon transition is 5.132, and this coefficient is positive and significant. The spillover effect of *ILU* on low-carbon transition for cities in the central region is also greater than 0 (coefficient of 3.824) but not significant. The results are different in the western region, where the spillover effect coefficient is negative and insignificant (−0.961). This result indicates that the *ILU* affects urban low-carbon transition differently depending on the geographical location. It plays an important driving role only in the eastern region.

**Table 9.** Results of spatial heterogeneity test.

**Panel A: Geographic Location.**

| Variable | UCR | UCR | UCR |
|---|---|---|---|
| | LR_Direct | LR_Indirect | LR_Total |
| East | 0.3455 ** | 5.1323 *** | 5.4778 *** |
| | (−1.962) | (−7.386) | (8.276) |
| Central | 1.6698 *** | 3.8238 | 5.4937 |
| | (−4.304) | (−1.093) | (1.534) |
| West | 0.5520 ** | −0.9605 | −0.4085 |
| | (−2.066) | (−0.435) | (−0.191) |

**Panel B: City Level.**

| Variable | UCR | UCR | UCR |
|---|---|---|---|
| | LR_Direct | LR_Indirect | LR_Total |
| Extra Large | 2.4806 *** | 7.9541 *** | 7.9253 *** |
| | (−3.913) | (−3.021) | (3.065) |
| Large | 0.1135 | 5.8359 *** | 5.9494 *** |
| | (−0.634) | (−4.801) | (4.901) |
| Moderate | 0.367 | −0.8509 | 1.6297 |
| | (−1.240) | (−0.657) | (1.395) |
| Small | −0.0288 | −7.4345 ** | −7.0676 ** |
| | (−0.078) | (−2.167) | (−2.026) |

**Panel C: City Circle.**

| Variable | UCR | UCR | UCR |
|---|---|---|---|
| | LR_Direct | LR_Indirect | LR_Total |
| Yangtze River Delta | 0.9410 * | 0.2061 | 1.1471 |
| | (−1.753) | (−0.108) | (0.625) |
| Beijing–Tianjin–Hebei City Circle | 5.3022 *** | 2.8549 | 8.1571 *** |
| | (−7.842) | (−0.960) | (2.619) |
| Middle Yangtze River Economic Belt | 2.8349 *** | −2.1027 | 0.7322 |
| | (−3.767) | (−0.754) | (0.251) |
| Pearl River Delta | −1.1533 *** | 2.7095 | 1.5562 |
| | (−3.996) | (−1.546) | (0.825) |
| Chengdu–Chongqing City Circle | 8.1728 *** | −4.4565 | 3.7163 |
| | (−5.553) | (−0.533) | (0.417) |

Note: The numbers in parentheses are robust t-statistics. ***, ** and * represent significance levels of 1%, 5% and 10%, respectively.

The cities are of different sizes. We use the year-end population of the city district as a proxy variable for city size. Given the frequent changes in the administrative divisions of city districts in many cities, we use the city size division criteria published by China in 2014 to select cities. Based on the total year-end population of city municipal districts, the 283 city samples can be categorized into four groups: extra-large cities (more than 5 million people), large cities (1 million to 5 million people), moderate cities (half a million to 1 million people), and small cities (less than half a million people). Among all the cities in the sample, there are 13 extra-large cities, 127 large cities, 98 moderate cities, and 45 small cities. As shown in Panel B of Table 9: The indirect effects for medium-sized and large cities are 7.954 and 5.836, respectively, and are significant at the 1% level. It indicates that the *ILU* of extra-large and large cities has a positive spillover effect on the low-carbon transition of neighboring cities. The indirect effect coefficient of moderate and small cities is negative, and small cities are significant at the 5% level. It implies that moderate and small

cities in China have caused some degree of pollution to the development of neighboring cities in promoting *ILU*. In addition, extra-large cities significantly promote local green transformation, with a coefficient value of 2.486.

We also test the spillover effects of *ILU* in different economic circles on urban low-carbon transformation separately according to the current policies of building urban circles implemented in China. Panel C of Table 9 shows that the *ILU* of the Yangtze River Delta, Beijing–Tianjin–Hebei urban circle, and Pearl River Delta cities form positive spillover to the low-carbon transition of neighboring cities. The middle reaches of the Yangtze River Economic Belt and Chengdu–Chongqing City Circle form a negative spillover. The coefficients of the indirect effects of the above five urban areas do not pass the significance test. It indicates that the *ILU* in China's urban areas did not significantly affect the low-carbon transition of neighboring cities in the promotion process. However, the *ILU* of the five urban areas all play a significant role locally. Only the *ILU* in the Pearl River Delta inhibits the local low-carbon transition, while the other four urban areas effectively promote the local low-carbon transition.

## 5. Discussion

In order to comply with the global low-carbon development trend, the Chinese government has proposed two significant goals of achieving "carbon peak" by 2030 and "carbon neutrality" by 2060, as well as the goals of promoting pollution reduction and carbon neutrality. Comprehensively promoting green and low-carbon transformation is an important strategic direction for economic and social development in the coming period. The baseline regression results in this paper verify that *ILU* has a significant positive spillover effect on low-carbon transformation, providing ideas for promoting *ILU* development to realize low-carbon urban transformation and carbon neutrality. This result is consistent with the findings of Shang et al. (2022) [56], but we go a step further by considering the influence of spatial factors and drawing conclusions about spillover effects and spatial boundaries. In addition, our findings further confirm the greenness and sustainability of China's *ILU* policy and urban spatial optimization [57]. China has implemented and is implementing integrated land use policies (e.g., Provisions on Saving and Intensive Land Use (2014) [16].As the spatial mainstay of economic development, land use intensification, while emphasizing the principles of "structural optimization" and "prioritizing conservation," has green, livable, and ecologically friendly connotations that contribute to the sustainable development of China's economy and the construction of an ecological civilization.

The existence of spatial boundaries provides some empirical evidence for local governments in China to break down inter-provincial administrative barriers and promote the articulation of urban spatial planning and ecological governance mechanisms between provinces. According to the division of China's administrative boundaries, 450 km is the average distance between China's provincial capital cities, and 250 km is the average radius of China's provinces. While the spillover effect of *ILU* on low-carbon city transformation can reach as far as 450 km, the peak of the spillover effect's coefficient is around 250 km. The driving force of *ILU* can reach as far as across China's provincial boundaries to affect the low-carbon urban transition in other provinces. However, local governments also need to be aware that the benefits are more significant in neighboring areas of the province.

The results on the heterogeneity of *ILU* dimensions and different spatial weights reflect the parts of the *ILU* policy promotion process that need attention and improvement. One of them is the need to pay attention to the goals of *ILU* policy promotion. According to the results of the four spatial weights, it can be found that *ILU* creates a driving effect on the low-carbon transition of neighboring and economically linked cities. It suggests cities which committed to promoting *ILU* can exert targeting, demonstration, and economic linkage effects. Considering that the driving effect of *ILU* is stronger in large cities, local governments can use large cities as their hinterland to play a "point-surface" driving role and thus promote the realization of the overall low-carbon transformation goals of Chinese cities. Second, it is essential to emphasize the means of *ILU* policies. Compared to ecological

carbon sink improvement and land use efficiency, the increase in land use density cannot significantly promote the local low-carbon transition. This result is similar to that of Baur et al. (2015) [58] based on data from European cities. Considering that land use density is related to the city's actual built-up area and the population's carrying capacity. This situation may arise because the current land intensification in China is still in the stage of capital intensification [59]. The most important vehicle for urban development is the construction land, and capital investment is concentrated in construction land. In this case, an increase in land use density will increase infrastructure and energy investment. Therefore, decisions about urban folding, spatial planning, and urbanization development need to be implemented prudently.

To promote China's low-carbon transition, paying attention to the spatial differences and regional cooperation in the environmental benefits of *ILU* policies is essential. This is consistent with the findings of current research in other countries [60,61]. First, it is necessary to pay attention to geographic location differences. The results of the heterogeneity regression show that only the cities in the eastern region can implement *ILU* policies while generating positive spillover effects on the low-carbon transition of neighboring cities, and the results in the central and western regions are not significant. In fact, the land carbon emissions of the eastern region, including Beijing, Tianjin, Shanghai, Jiangsu Province, Zhejiang Province and other provinces, account for a high proportion of the national emissions (see Figure 1); however, eco-efficiency and energy use efficiency are also higher in the eastern coastal region than in the central and western regions [62,63]. Taken together, although the eastern region faces more substantial pressure to reduce emissions, it has developed a more inclusive green land use system over the years and has experience in urban spatial planning and low-carbon environmental management, thus creating a demonstration effect on neighboring cities. These achievements may be related to the greater concentration of talent, technology, and innovation in the eastern region, which could receive further attention. On the contrary, the central and western regions, such as Sichuan, Hubei, Henan, and other provinces, are all in rapid economic development, with large populations and high pressure on land use carbon emissions. Promoting *ILU* in cities in the central and western regions is more challenging and requires more advanced experience from the eastern regions. Secondly, it is necessary to pay attention to urban-level differences. The positive spillover effect of *ILU* on urban low-carbon transition is not evident in small and medium-sized cities and urban circles. These findings reflect two aspects: small and medium-sized cities have weaker governance capacity and may face more difficult ecological governance and spatial layout adjustment [57]. The second is that administrative barriers within the city region have not yet been broken down [64], and there is less willingness to cooperate between city clusters [65], which limits the direct and spillover effects of *ILU* policies in the city region on the low-carbon transition. The above conclusions provide ideas for future synergistic promotion of low-carbon transformation in Chinese cities.

Although this study complements the shortcomings of the studies related to *ILU* and urban low-carbon transition, it also provides a theoretical reference for the study of urban low-carbon transition at the spatial level. However, there are still certain shortcomings in this study that need to be improved. First, the low-carbon transition of cities mainly includes two components of low-carbon and economic development. In the future, we will continue exploring the low-carbon land use in the low-carbon transition of cities and further enrich the urban environment-related research. Second, limited by data availability, the measurement of *ILU* still needs to be improved. Meanwhile, the connotation of *ILU* is constantly being improved and has likewise changed regarding green and blue infrastructure construction. We expect to expand on these two aspects in the subsequent study to improve the measure of *ILU*. Third, the research questions and scope can be extended to other developing countries.

**6. Conclusions and Policy Implications**

*6.1. Conclusions*

This study incorporates China's *ILU* policy and low-carbon transition into a unified analytical framework from the perspective of spatial spillover. The spatial autocorrelation, evolution, spillover effects, spatial decay, and spatial heterogeneity of the intensive land use construction in China's low-carbon transition are investigated based on theoretical and empirical analyses. Many valuable conclusions are drawn as follows:

(1) intensive land use has significant spatial spillover effects on the low-carbon transition of cities. It not only plays a positive role for neighboring cities but also promotes the low-carbon transformation of economically related cities. The exogenous shock test of the pilot cities of new urbanization also verifies this result. Furthermore, the spillover effect of new urbanization exists in the range of 0–450 km and peaks at about 250 km.

(2) The results of mechanism validation indicate that industry transfer and technology spillover are dual mechanism channels for intensive land use for low-carbon transition in China.

(3) intensive land use plays a catalytic role in the low-carbon transition of surrounding cities through three dimensions: the density of land use, economic and social efficiency, and ecology benefits, among which the ecological dimension has the most potent effect. In contrast, land use density does not contribute to the local low-carbon transition.

(4) As an essentially urban development strategy, the intensive land use in the eastern region and large-sized cities can significantly contribute to the low-carbon transition of neighboring cities. However, intensive land use in urban economic zones cannot perform the radiation effect to drive the low-carbon transformation of neighboring cities.

On average, the obtained results are broadly consistent with the existing literature. At the same time, it provides more novel and convincing evidence on the effectiveness of intensive land use and the importance of clean technology development and production transformation for carbon neutrality in emerging countries such as China. It will help promote Chinese government departments to promote and implement *ILU* policies and action plans for dual carbon goals in a targeted manner and also help provide theoretical and empirical references for developing countries to promote environmentally friendly and coordinated development.

*6.2. Policy Implications*

We propose the following policy recommendations mainly from the perspective of developing countries choosing governments as facilitators.

First, the government should increase *ILU* through specific measures to ensure that it plays an active role in the low-carbon transition of the city. The government needs to focus on formulating economic, social, and ecological policies. Quality improvement should not be neglected in favor of economic growth, and excessive "accumulation" of factors of production on land will ultimately lead to lower efficiency and higher emissions. At the same time, the government should control the increase of land and explore the land stock, coordinate the safeguarding of development and the protection of resources, optimize the allocation of factors through industrial upgrading, strengthen the control of urban growth boundaries, rationally allocate land resources, and enhance the intensive level of urban land use.

Second, the government needs to use the technical means of regional integration to optimize the intensive use of urban land in the city circle. Integrating national economic and social development planning, land use master planning, land improvement planning, area development planning, ecological environment planning, and enterprise access catalogs are important designs worth trying by local governments. This top-level design can strengthen the long-term mechanism of land use optimization under the control and integration of multiple regulations and cultivate the concept of regional integration. As a result, it will enhance the synergistic and consistent mechanism for optimizing the allocation of land resources.

Thirdly, the local government should use land-space constraints on economic activities to promote a cleaner transformation of the industrial structure and the process of promoting green technologies. Transforming the existing unreasonable and unclean industrial structure is essential for developing an intensive and efficient economic growth mode. On the one hand, local governments need to protect patented technologies and promote enterprises' green technology research and development process. On the other hand, the government needs to promote the cleanliness of the energy structure and strengthen the rational allocation of industrial development and energy structure. In addition, the government should play an organizing and guiding role in promoting green and healthy production methods such as recycling, high efficiency, and emission reduction and improve the city's scientific and technological innovation capacity and technology spillover effect. In this way, a virtuous green circular economy can be formed within and among cities.

Finally, local governments should choose and implement *ILU* policy tools according to local conditions. The government needs to form a differentiated idea of intensive land use. Specifically, the government can build a differentiated control system for intensive land use based on the zoning of land intensive use evaluation results, zoning of resource and environmental carrying status, and zoning of dominant industries, and from the perspective of economic development priority and resource and environmental utilization and protection priority.

**Author Contributions:** Conceptualization, X.L.; methodology, Y.G.; software, G.W.; validation, X.L.; formal analysis, X.L.; investigation, X.L.; resources, Y.G.; data curation, X.L.; writing—original draft preparation, X.L.; writing—review and editing, G.W.; visualization, Y.G.; supervision, G.W.; project administration, Y.G.; funding acquisition, Y.G. All authors have read and agreed to the published version of the manuscript.

**Funding:** This research was funded by the National Social Science Foundation of China, grant number: 20BJL142, Postdoctoral Science Foundation of China, grant number: 2022M720131, Key Research and Special Popularization (Soft Science Research) Projects of Henan Province in 2022, grant number: 222400410135, and Carbon Emissions Trading Provincial and Ministerial Joint Construction Collaborative Innovation Project Center of Hubei Province, grant number 22CICETS-YB011.

**Data Availability Statement:** The data are available on request.

**Acknowledgments:** Thanks for the support from Hubei University and Guizhou University.

**Conflicts of Interest:** The authors declare no conflict of interest.

## Appendix A

**Table A1.** Summary table of acronyms.

| Acronyms | Description |
| --- | --- |
| *LCT* | Low-carbon transition |
| *ILU* | Intensive land use |
| *ER* | Environmental regulation |
| *TEM* | annual average temperature |
| *OPEN* | Openness to foreign investment |
| *GOV* | Government Intervention |
| *AGG* | Industrial agglomeration |
| *MAK* | Marketization |
| *FIAN* | Financial development |
| *IS* | Industrial structure transformation |
| *TS* | Technology spillover |

**Table A2.** Results of spatial autocorrelation test.

| Year | Moran's I | | Geary's C | |
|---|---|---|---|---|
| 2006 | 0.076 *** | (3.137) | 0.919 *** | (−2.697) |
| 2007 | 0.082 *** | (3.359) | 0.912 *** | (−2.942) |
| 2008 | 0.117 *** | (4.765) | 0.880 *** | (−4.013) |
| 2009 | 0.124 *** | (5.035) | 0.879 *** | (−4.074) |
| 2010 | 0.125 *** | (5.049) | 0.874 *** | (−4.241) |
| 2011 | 0.138 *** | (5.558) | 0.855 *** | (−4.890) |
| 2012 | 0.127 *** | (5.160) | 0.860 *** | (−4.661) |
| 2013 | 0.133 *** | (5.358) | 0.863 *** | (−4.674) |
| 2014 | 0.099 *** | (4.040) | 0.904 *** | (−3.258) |
| 2015 | 0.134 *** | (5.399) | 0.863 *** | (−4.696) |
| 2016 | 0.164 *** | (6.596) | 0.843 *** | (−5.500) |
| 2017 | 0.061 ** | (2.535) | 0.962 | (−1.353) |
| 2018 | 0.158 *** | (6.351) | 0.850 *** | (−5.250) |
| 2019 | 0.202 *** | (8.097) | 0.816 *** | (−6.303) |

Notes: *** $p < 0.01$, ** $p < 0.05$. Z-values in parentheses.

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
