# Peer review of "How Does Intensive Land Use Affect Low-Carbon Transition in China? New Evidence from the Spatial Econometric Analysis"

_land, doi:10.3390/land12081578_

Round 1

Reviewer 1 Report

This paper uses urban data from China to examine the association between intensive land use and low-carbon transition. The results show a positive association between the two even after controlling for various factors.

I really like the originality of this paper because low carbon transition is an important research topic involving the future economic development of economies around the world. Due to the rapid growth of the Chinese economy, the pressure for a low-carbon transition in China is increasing. The question of how the land can be utilized in a low-carbon manner is of outstanding value in Chinese-style "land finance."

However, I also have many questions and concerns:

1.         It is not clear to me why the authors analyzed the "ecological" and "economic" systems first in the theoretical analysis but did not explicitly mention these two systems in the subsequent spillover analysis. I suggest the authors add visualization to express the relationship between intensive land use and low-carbon economic transformation.

2.         In the abstract, the authors summarize that land intensification has essential implications for old urban areas. In the following test, the authors analyze China's critical urban economic circles. It is somewhat confusing. I suggest that the authors provide a more detailed analysis. To be precise, old urban areas and urban processes are different geographical concepts.

3.         In section 4.2, the authors conduct a robustness test using the "new urbanization" policy as the exogenous shock variable. It is creative. According to the authors, land intensification is one of the core aspects of the new urbanization policy. I suggest adding explanations in this section to rationalize the choice of variables.

4.         In Subsection 4.3.4, the measurement of channel variables lacks theoretical basis and literature support.

5.         The paper requires professional native English editing. Please write your text in good and concise English.

Here are some minor issues:

1.         The analysis of spatial autocorrelation is of some value, but the table of tests can be placed in the appendix.

2.         Some redundant examples in the introduction could be deleted. For example, in the first paragraph, too much explanation about the importance.

 The paper requires professional native English editing. Please write your text in good and concise English.

Author Response

Response to Reviewer #1:

We would like to thank the reviewer for your careful review and comments on our manuscript. Your comments help us improve the quality of our work. In your comments, you pointed out accurately the existing shortcomings we ignored, and also gave the practical solutions. We are honored to receive your approval of the value of this manuscript. We agree and have made revisions according to each of your comments. Our responses are organized point by point based on your comments.

Reviewer #1:

This paper uses urban data from China to examine the association between intensive land use and low-carbon transition. The results show a positive association between the two even after controlling for various factors.

I really like the originality of this paper because low carbon transition is an important research topic involving the future economic development of economies around the world. Due to the rapid growth of the Chinese economy, the pressure for a low-carbon transition in China is increasing. The question of how the land can be utilized in a low-carbon manner is of outstanding value in Chinese-style "land finance. 

However, I also have many questions and concerns.

Response:

Thank you very much for your kind and positive comments above. We have done our best to revise the manuscript, but if any additional revision is needed, we will certainly do so under your directions.

Comments 1:

It is not clear to me why the authors analyzed the "ecological" and "economic" systems first in the theoretical analysis but did not explicitly mention these two systems in the subsequent spillover analysis. I suggest the authors add visualization to express the relationship between intensive land use and low-carbon economic transformation.

Response:

Thank you for your valuable comments and suggestions. We are very sorry to have caused you such distress. We look at the direct impact of land intensification on low-carbon transition through the dual dimensions of ecology and economy. Still, spillover effects are viewed from a spatial perspective, where the ecological and economic dimensions are inextricably intertwined. This paper focuses on explaining neighboring cities and economically connected cities in the spillover effect section, and they embody both environmental and economic dimensions. Based on your suggestions, we have revised the Mechanism and Research Hypothesis (Section 2). However, we did not add a visualization of this section because our mapping of the mechanisms was not very effective and it was difficult to clearly show the relationship between land intensification and low carbon transition.

The revised sections are as follows:

Revision

2. Mechanism and Research Hypothesis

…Economic green transformation is the primary way to realize the goal of sustainable development, focusing on economic efficiency and carbon reduc-tion [33]. Compared with traditional land use, ILU embodies the concept of scientific and clean development. Whether it is to reduce emission sources or increase carbon sink ab-sorption, ILU plays an important role[34]. The mechanism between ILU and China's low-carbon transition can be analyzed in economic and ecological systems.

2.1. Promoting effect of intensive land use on low-carbon transition

First, we consider the economic growth effect. ILU is to increase the input of factors such as capital, labor, and technology on the urban land stock and improve land use efficiency through rational layout and optimization of land use structure to promote sustainable development [35]. According to the law of increasing and decreasing land remuneration, before reaching the highest point of remuneration, the more capital and labor force invested in the land per unit area, the higher the economic output obtained, that is, the higher the intensity of land use and development, the higher the contribution to eco-nomic growth [36].

Second, we consider the carbon emission reduction effect. At the ecosystem level, ILU corresponds to the impact of land use change on soil carbon stock and vegetation carbon stock. According to the land use classification, construction land is the primary carbon source, while ecological and agricultural land are essential sources of carbon sinks. ILU effectively reduces the conversion of agricultural and ecological land, such as garden land, forest land, and grassland, to construction land and increases carbon sink absorption within the ecosystem [37]. In addition, at the economic system level, with the strengthening of land use constraints, low energy-consuming technologies, enterprises and industrial chains will be "squeezed out", thus reducing carbon emissions. Compact land space pattern facilitates public transportation use and reduces infrastructure construction waste [38]. It helps to improve the efficiency of centralized energy supply and utilization and reduces the growth of carbon sources from land construction.

2.2 Spatial Spillover Mechanism

According to the theory of agglomeration effect, the increased density and spatial proximity of economic activities on land contribute to the economies of scale in production and transactions and resulting spillovers on a local scale [39]. Considering that car-bon emissions are mainly influenced by socioeconomic drivers such as the stage of eco-nomic development, energy resource endowment, and consumption patterns. Land use mode, scale, structure, and intensity are closely related to industrial development status and technological progress [40]. Therefore, intensive land use affects the low-carbon trans-formation of neighboring cities mainly by influencing the socioeconomic constitutive fac-tors reflected in industrial development and technological progress [41][42].

The mediating role of industrial structure transformation. On the one hand, as the land use pattern constrains the industrial layout, the crude land approach is challenging to promote the automatic transformation of industry. The theoretical connotation of ILU includes the principles of "structural optimization" and "market allocation." Conversely, ILU encourages the development of resource-saving and environment-friendly industries, which helps cities establish a modernized industrial system characterized by low energy consumption and emissions [40]. On the other hand, due to market competition and the price mechanism, industrial agglomeration forces enterprises to improve production technology and techniques and promotes upgrading the industrial chain layout [43]. As a result, with the gradual withdrawal of resource-intensive enterprises, the utility of energy utilization has been enhanced. The booming development of high-tech enterprises and service industries is conducive to easing resource dependence and environmental pressure. This transformation of industrial structure promotes the diversified division of labor among enterprises, which helps the low-carbon production links and industrial clusters among cities to generate symbiosis, complement each other, and promote the low-carbon transformation and development of neighboring cities [44].

The intermediary role of technology spillover. The agglomeration economy formed by ILU makes cities accumulate innovative resources and also promotes technological over-flow from cities. The theory of ILU includes the principles of "conservation first" and "re-form and innovation," which help accelerate the R&D, innovation, and application of production, environmental protection, and energy-saving technologies and generate technological factor overflows from the region to neighboring or related regions. The overflow accelerates the learning and disseminating of green technologies and innovation among cities. Neighboring cities at this time can realize imitation and secondary innovation with the help of technological overflow, thus promoting green transformation.

Overall, spatial spillovers from economic systems and ecosystems diffuse the effects of NTUs on local industrial restructuring and technological spillovers to neighboring regions, which ultimately manifests itself in the diffusion of local ILU's driving effects on neighboring regions' LCT. The above spillover mechanisms accumulate layer by layer, forming the total effect of ILU on the overall regional LCT.

2.3 Spatial Decay Mechanism

Most studies recognize that spillover effects are characterized by spatial decay. It is because the cost of logistics and information exchange will rise with increased geographical distance and the restriction of administrative boundaries. As a result, the spatial spillover effect of ILU may show a specific attenuation pattern and boundary effect as the possibility of factor spillover decreases.

First, we consider the role of geographic distance. Studies have confirmed that information dissemination shows the law of attenuation with increased geographical distance. As the distance between neighboring cities and knowledge-center cities increases, the efficiency of information dissemination decreases [45]. Furthermore, increased geographic distance leads to interfirm transaction costs, transportation costs, and risk control. This former is a function of the geographic distance between the cooperating parties [46,47], and the latter is reflected in the increased risk of breach of contract and the reduced level of trust, among others. These are not conducive to inter-city technology, exchange and cooperation, leading to spatial limitations in the spillover effects of ILU on low-carbon transitions in neighboring cities.

For specific revisions, please refer to the line 145-230 on page 5-6 of the Revised Manuscript (with changes marked).

Comments 2:

In the abstract, the authors summarize that land intensification has essential implications for old urban areas. In the following test, the authors analyze China's critical urban economic circles. It is somewhat confusing. I suggest that the authors provide a more detailed analysis. To be precise, old urban areas and urban processes are different geographical concepts.

Response:

Thank you for pointing this out and providing kind advice. According to the heterogeneity test results, the environmental effect of land intensification can be better exerted within the city cluster. Still, the city cluster has a weaker radiation power than the outside. Moreover, only city clusters with a more extended development history positively affect land intensification. Therefore, we briefly used the word "old" in the abstract to reflect this feature we found, but, unfortunately, this word is not accurate. We apologize for the confusion. In summary, We replaced "old urban areas" with " veteran economic circles."

The revised sections are as follows:

Revision

Abstract: …the study provides empirical evidence that intensive land use can significantly promote low-carbon transition in neighboring and economically linked cities (especially in eastern cities, large and medium-sized cities, and veteran economic circles)…

For specific revisions, please refer to the line 22 on page 1 of the Revised Manuscript (with changes marked).

Comments 3:

In section 4.2, the authors conduct a robustness test using the "new urbanization" policy as the exogenous shock variable. It is creative. According to the authors, land intensification is one of the core aspects of the new urbanization policy. I suggest adding explanations in this section to rationalize the choice of variables.

Response:

Thank you for the useful suggestions and your point is well-taken. We have added a detailed explanation of the use of the indicator "new urbanization policies" in Section 3.2.3 to further rationalize the choice of variables by further justifying the relationship between intensive land use and new urbanization policies. The revised sections are as follows:

Revision

3.2.3. Spatial difference-in-differences model

According to the guideline of the Ministry of Land and Resources of China, the eco-nomic and intensive use of land is a strategic choice for new urbanization. Land-use intensification is an important policy tool in promoting the implementation of new urbanization. Therefore, this paper considers the new urbanization contention policy a proxy variable for ILU. In this study, we …

For specific revisions, please refer to the line 306-309 on page 8 of the Revised Manuscript (with changes marked).

Comments 4:

In Subsection 3.3.4, the measurement of channel variables lacks theoretical basis and literature support.

Response:

Thank you for your constructive comments on variable measurement. We respond as follows:

We only introduced how the two mechanism variables work in Section 2 “Mechanism and Research Hypothesis” and did not introduce them in detail, which is our negligence. Following your suggestion, we added the basis and literature support for variable measurement in Subsection 3.3.4. The revised sections are as follows:

Revision

3.3.4. Other variables

2. Channel variables

Based on the theoretical analysis in the previous subsections, the two key mechanism variables for channel analysis are industrial structure transformation (IS) and technology spillover (TS).

Industrial structure transformation (IS). The upgrading of industrial structure towards cleanliness is the key to realizing the goal of green development. The current indicators for industrial structure upgrading mainly use internal structure change, energy consumption per unit GDP of industry, and product sales of pollution-intensive industries. We utilize the entropy value method to determine the degree of cleaner transformation of industrial structure. In this paper, we refer to Zhang et al. (2023) to construct the indicator system from two aspects of clean energy consumption and clean production. Clean energy consumption is measured by the ratio of total industrial energy consumption to industrial added value; clean production is expressed by the ratio of regional industrial added value to carbon emissions. Through the dimensionless quantization of the indicators, the entropy value method is then used to identify the degree of cleaner transformation of the industrial structure.

Technology spillover (TS). Generally speaking, due to China's imperfect patent guarantee mechanism and relatively backward R&D capability, it is difficult for enterprises to convert R&D inputs into green innovation outputs. In contrast, the number of green patents can reflect the actual innovation outputs more objectively. We use the number of green patent acquisitions obtained in one year as a proxy variable for technology spillovers. 

For specific revisions, please refer to the line 408-425 on page 11-12 of the Revised Manuscript (with changes marked).

Comments 5:

The paper requires professional native English editing. Please write your text in good and concise English.

Response:

Thanks for your suggestions, and your point is well-taken. We apologize for the problems with the language. The language of the paper has been carefully checked and corrected with the assistance of native English-speaking experts. We hope that language fluency has been substantially improved. If our revision is not up to your standard yet, we would be happy to revise it again.

Comments 6:

The analysis of spatial autocorrelation is of some value, but the table of tests can be placed in the appendix.

Response:

Thank you for your useful comments and your point is well-taken. Based on the kind suggestion, we have removed the Table 4 (Results of spatial autocorrelation test.)  from Section 4.1.1, and placed it in the table at the end of the manuscript (see Appendix A.2).

Please refer to the line 1031-1033 on page 28 of the Revised Manuscript (with changes marked).

Comments 7:

Some redundant examples in the introduction could be deleted. For example, in the first paragraph, too much explanation about the importance.

Response:

Thank you for the valuable suggestions. We only retained the necessary explanations of the importance of the low-carbon transition. For visualization, we added a map of the spatial distribution of carbon emissions from land use in each of China's provinces. The revised sections are as follows.

Revision

1. Introduction

Global warming caused by greenhouse gas (GHG) emissions seriously threatens the natural and social environments on which human beings depend for survival[1,2]. The series of chain reactions across ecosystems triggered by greenhouse gases has become a massive challenge for all humanity[3,4]. The International Energy Agency (IEA) estimates that global energy-related carbon dioxide (CO2) emissions will grow by 0.9% in 2022, reaching a record high of over 36.8 Gt[5]. Among them, carbon dioxide emissions from energy combustion and industrial processes account for 89% of total energy-related greenhouse gas emissions; methane from energy combustion, leakage, and venting ac-counts for 10%. They are all mainly from onshore oil and gas field operations and the production of coal for power. Compared to 1880, 2022 is also the fifth hottest year globally, fraught with extreme weather events[6,7].As the country with the most rapid economic development in the 20th century, China has become the world's largest emitter of carbon dioxide since 2007,, with carbon emissions rising from 8.83 billion tons in 2011 to 9.90 billion tons in 2020[8]. As the urbanization rate of the population rises (to 64.72% in 2021), large-scale migration and the concentration of human activities will result in continued land expansion and land carbon emissions…

Please refer to the line 39-44 on page 1 of the Revised Manuscript (with changes marked).

Reviewer 2 Report

This paper creatively incorporates intensive land use and low-carbon transition into a research framework with strong scientific research significance. Using municipal panel data, the relationship between intensive land use and low-carbon transition is carefully studied using Spatial Durbin Model, which is attractive. The framework and content of the paper have good integrity, but it still has some shortcomings, and it is expected that the authors will make meticulous revisions:

1. It can be seen that this paper has a novel entry point, please focus on describing the marginal contribution, highlighting the innovation, and increasing the readability of this paper.

2. The elaboration of the theoretical mechanism should be logical and universal. As far as possible, please clarify the theoretical mechanism of the paper and compare it with the existing literature to emphasize the feasibility of the theoretical mechanism of this paper.

3. This paper utilizes the Spatial Difference in Difference model to conduct relevant empirical research. Before utilizing the Difference in Difference model to assess the policy effect you should first conduct a stability test of the panel data to illustrate the correctness of the use of the model through the placebo and parallel trend tests. This point needs to be carefully considered clearly by the authors.

4. The empirical results of this paper should be fully compared with the existing related literature so as to illustrate the scientificity and theoretical of the results of this study. The findings and novelty of this paper are highlighted side by side.

5. Policy recommendations should be formulated based on the findings of the study. The authors are requested to scrutinize this point carefully so that the policy recommendations and the research findings correspond to each other.

Author Response

Response to Reviewer #2:

We would like to thank the reviewer for your careful review and comments on our manuscript. Your comments help us improve the quality of our work. In your comments, you pointed out accurately the existing shortcomings we ignored, and also gave the practical solutions. We are honored to receive your approval of the value of this manuscript. We agree and have made revisions according to each of your comments. Our responses are organized point by point based on your comments.

Reviewer #2:

This paper creatively incorporates intensive land use and low-carbon transition into a research framework with strong scientific research significance. Using municipal panel data, the relationship between intensive land use and low-carbon transition is carefully studied using Spatial Durbin Model, which is attractive. The framework and content of the paper have good integrity, but it still has some shortcomings, and it is expected that the authors will make meticulous revisions.

Response:

Thank you very much for your kind and positive comments above. We have done our best to revise the manuscript, but if any additional revision is needed, we will certainly do so under your directions.

Comments 1:

It can be seen that this paper has a novel entry point, please focus on describing the marginal contribution, highlighting the innovation, and increasing the readability of this paper.

Response:

Thank you for your valuable comments and suggestion. We have supplemented the interpretation of the the marginal contribution and carefully reviewed the introduction and made changes and additions.

The revised sections are as follows:

Revision

1.Introduction

Based on the consideration of breaking through the limitations of the existing literature, we decided to identify and assess the driving effect of land intensification on China's low-carbon transition from a spatial perspective and evaluate the mechanism of its action in terms of both green upgrading of industries and clean technology spillovers (Figure 1). The possible marginal contribution consists of the following three points. Firstly, we constructed a framework for the relationship between land use intensification and low carbon transition considering both direct and spatial driving effects and numericized land use intensification and low carbon transition in the form of multiple composite indicators. Compared with traditional studies, we provide a more comprehensive analysis from a spatiotemporal perspective (i.e., spatial distribution, spatial autocorrelation, evolution, spillover effects, spatial decay, and spatial heterogeneity). Secondly, compared with the traditional single research method, we adopt the exogenous policy shock test to support the conclusion of the driving effect and examine the dual channels of influence of land intensification and low-carbon transition from the dual perspectives of green transformation of industrial structure and clean technology spillovers, which expands the empirical research in related fields. Finally, we provide practical policy recommendations for policymakers regarding the efficiency of low-carbon economies and the focus on green, livable, and efficient living and production environments in emerging countries such as China.

For specific revisions, please refer to the line 123-141 on page 3-4 of the Revised Manuscript (with changes marked).

Comments 2:

The elaboration of the theoretical mechanism should be logical and universal. As far as possible, please clarify the theoretical mechanism of the paper and compare it with the existing literature to emphasize the feasibility of the theoretical mechanism of this paper.

Response:

Thank you for your constructive comments on the theoretical mechanism. According to your suggestion, we compare it with the existing literature to emphasize the feasibility of the theoretical mechanism of this paper. The literature added is shown below:

[1] Lee, C.C.; Feng, Y.; Peng, D. A Green Path towards Sustainable Development: The Impact of Low-Carbon City Pilot on Energy Transition. Energy Economics 2022, 115, 106343, doi:10.1016/j.eneco.2022.106343.

[2] Feng, H.; Wang, S.; Zou, B.; Yang, Z.; Wang, S.; Wang, W. Contribution of Land Use and Cover Change (LUCC) to the Global Terrestrial Carbon Uptake. Science of The Total Environment 2023, 901, 165932, doi:10.1016/j.scitotenv.2023.165932.

[3] Chuai, X.; Xia, M.; Ye, X.; Zeng, Q.; Lu, J.; Zhang, F.; Miao, L.; Zhou, Y. Carbon Neutrality Check in Spatial and the Response to Land Use Analysis in China. Environmental Impact Assessment Review 2022, 97, 106893, doi:10.1016/j.eiar.2022.106893.

[4] Popp A.; Humpenöder F.; Weindl I.; Bodirsky B.L.; Bonsch M.; Lotze-Campen H.; Müller C.; Biewald A.; Rolinski S.; Stevanovic M.; et al. Land-use protection for climate change mitigation. Nature Climate Change 2014, 4, 1095–1098.

[5] Nichols, D.A. Land and economic growth. Amer. econ. Rev., 1970, 60, 332–340.

[6] Shao, Q.; Zhang, W.; Cao, X.; Yang, J.; Yin, J. Threshold and Moderating Effects of Land Use on Metro Ridership in Shenzhen: Implications for TOD Planning. Journal of Transport Geography 2020, 89, 102878, doi:10.1016/j.jtrangeo.2020.102878.

[7] Yao, Y.; Pan, H.; Cui, X.; Wang, Z. Do Compact Cities Have Higher Efficiencies of Agglomeration Economies? A Dynamic Panel Model with Compactness Indicators. Land Use Policy 2022, 115, 106005, doi:10.1016/j.landusepol.2022.106005.

[8] Dong, Y.; Jin, G.; Deng, X. Dynamic Interactive Effects of Urban Land-Use Efficiency, Industrial Transformation, and Carbon Emissions. Journal of Cleaner Production 2020, 270, 122547, doi:10.1016/j.jclepro.2020.122547.

[9]Wang, F.; Yang, F. A Review of Research on China’s Carbon Emission Peak and Its Forcing Mechanism. Chinese Journal of Population Resources and Environment 2018, 16, 49–58, doi:10.1080/10042857.2018.1433810.

[10] Chen, W.; Shen, Y.; Wang, Y.; Wu, Q. The Effect of Industrial Relocation on Industrial Land Use Efficiency in China: A Spatial Econometrics Approach. Journal of Cleaner Production 2018, 205, 525–535, doi:10.1016/j.jclepro.2018.09.106.

[11] Peng, C.; Song, M.; Han, F. Urban Economic Structure, Technological Externalities, and Intensive Land Use in China. Journal of Cleaner Production 2017, 152, 47–62, doi:10.1016/j.jclepro.2017.03.020.

[12] Liu, X.; Zhang, W.; Cheng, ·Jing; Zhao, S.; Zhang, X. Green Credit, Environmentally Induced R&D and Low Carbon Transition: Evidence from China. Environ Sci Pollut Res 2022, 29, 89132–89155, doi:10.1007/s11356-022-21941-0.

For specific revisions, please refer to the line 879-1020 on page 25-27 of the Revised Manuscript (with changes marked).

Comments 3:

This paper utilizes the Spatial Difference in Difference model to conduct relevant empirical research. Before utilizing the Difference in Difference model to assess the policy effect you should first conduct a stability test of the panel data to illustrate the correctness of the use of the model through the placebo and parallel trend tests. This point needs to be carefully considered clearly by the authors.

Response:

Thank you for your valuable suggestions. The standardized policy effects assessment process should include stability tests, placebo tests, etc., as you point out. However, the purpose of using policy effects assessment in this paper is to provide a robust test of the environmental impact of intensive land use. More analysis of the policy assessment would clarify this paper's focus. It is not appropriate to put these test results into the main text. Therefore, to ease your concerns, we would like to report the results of this part of the construction to you in the hope of obtaining your approval of the methodology and test results. However, we have not included them in the text.

Parallel trend test and placebo test plots are shown below:

Comments 4:

The empirical results of this paper should be fully compared with the existing related literature so as to illustrate the scientificity and theoretical of the results of this study. The findings and novelty of this paper are highlighted side by side.

Response:

Thank you for your constructive comments on the empirical results. According to your suggestion, we compare it with the existing literature to emphasize the feasibility of the empirical results of this paper. The literature added is shown below:

Revision

5.Discussion

…The baseline regression results in this paper verify that ILU has a significant positive spillover effect on low-carbon transformation, providing ideas for promoting ILU development to realize low-carbon urban transformation and carbon neutrality. This result is consistent with the findings of Shang et al. (2022) [56], but we go a step further by considering the influence of spatial factors and drawing conclusions about spillover effects and spatial boundaries. In addition, our findings further confirm the greenness and sustainability of China's ILU policy and urban spatial optimization [57]. China has implemented and is implementing integrated land use policies (e.g., Provisions on Saving and intensive Land Use (2014) [16]. As the spatial mainstay of …

…Considering that the driving effect of ILU is stronger in large cities, local governments can use large cities as their hinterland to play a "point-surface" driving role and thus promote the realization of the overall low-carbon transformation goals of Chinese cities. Second, it is essential to emphasize the means of ILU policies. Compared to ecological carbon sink improvement and land use efficiency, the increase in land use density cannot significantly promote the local low-carbon transition. This result is similar to that of Al-bert et al. (2015) [58] based on data from European countries. Considering that land use density is related to the city's actual built-up area and the population's carrying capacity. This situation may arise because the current land intensification in China is still in the stage of capital intensification [59]. The most important vehicle for urban development is the construction land, and capital investment is concentrated in construction land. In this case, an increase in land use density will increase infrastructure and energy investment. Therefore, decisions about urban folding, spatial planning, and urbanization development need to be implemented prudently.

To promote China's low-carbon transition, paying attention to the spatial differences and regional cooperation in the environmental benefits of ILU policies is essential. This is consistent with the findings of current research in other countries [60][61]. First, it is necessary to pay attention to geographic location differences. The results of the heterogeneity regression show that only the cities in the eastern region can implement ILU policies while generating positive spillover effects on the low-carbon transition of neighboring cities, and the results in the central and western regions are not significant. In fact, the land carbon emissions of the eastern region, including Beijing, Tianjin, Shanghai, Jiangsu Province, Zhejiang Province, etc., account for a high proportion of the national emissions (see Figure 1); however, eco-efficiency and energy use efficiency are also higher in the east-ern coastal region than in the central and western regions [62][63]. Taken together, although the eastern region faces more substantial pressure to reduce emissions, it has developed a more inclusive green land use system over the years and has experience in urban spatial planning and low-carbon environmental management, thus creating a demonstration effect on neighboring cities. These achievements may be related to the greater concentration of talent, technology, and innovation in the eastern region, which could receive further attention. On the contrary, the central and western regions, such as Sichuan, Hubei, Henan, and other provinces, are all in rapid economic development, with large populations and high pressure on land use carbon emissions. Promoting ILU in cities in the central and western regions is more challenging and requires more advanced experience from the eastern regions. Secondly, it is necessary to pay attention to urban level differences. The positive spillover effect of ILU on urban low-carbon transition is not evident in small and medium-sized cities and urban circles. These findings reflect two aspects: small and medium-sized cities have weaker governance capacity and may face more difficult ecological governance and spatial layout adjustment [57]. The second is that administrative barriers within the city-region have not yet been broken down [64], and there is less willingness to cooperate between city clusters [65], which limits the direct and spillover effects of ILU policies in the city-region on the low-carbon transition. The above conclusions provide ideas for future synergistic promotion of low-carbon transformation in Chinese cities.

The literature added is shown below:

  1. Shang, Y.; Xu, J.; Zhao, X. Urban Intensive Land Use and Enterprise Emission Reduction: New Micro-Evidence from China towards COP26 Targets. Resources Policy2022, 79, 103158, doi:10.1016/j.resourpol.2022.103158.
  2. Li, X.; Wang, L. Does Administrative Division Adjustment Promote Low-Carbon City Development? Empirical Evidence from the “Revoke County to Urban District” in China. Environ Sci Pollut Res2023, 30, 11542–11561, doi:10.1007/s11356-022-22653-1.
  3. Baur, A.H.; Förster, M.; Kleinschmit, B. The Spatial Dimension of Urban Greenhouse Gas Emissions: Analyzing the Influence of Spatial Structures and LULC Patterns in European Cities. Landscape Ecol2015, 30, 1195–1205, doi:10.1007/s10980-015-0169-5.
  4. Wu, Y.; Ren, Y.; Xu, Z. Market allocation of land elements under the national spatial planning system: theory, mechanism and model. China Land Sciences2023, 37, 28–37.
  5. Bridge, G.; Bouzarovski, S.; Bradshaw, M.; Eyre, N. Geographies of Energy Transition: Space, Place and the Low-Carbon Economy. Energy Policy2013, 53, 331–340, doi:10.1016/j.enpol.2012.10.066.
  6. Hill, D. Regional Cooperation and Asia’s Low Carbon Economy Transition: The Case of New Zealand. In Investing on Low-Carbon Energy Systems: Implications for Regional Economic Cooperation; Anbumozhi, V., Kalirajan, K., Kimura, F., Yao, X., Eds.; Springer: Singapore, 2016; pp. 309–326 ISBN 978-981-10-0761-3.
  7. Zhang, Y.; Wang, L.; Tang, Z.; Zhang, K.; Wang, T. Spatial Effects of Urban Expansion on Air Pollution and Eco-Efficiency: Evidence from Multisource Remote Sensing and Statistical Data in China. Journal of Cleaner Production2022, 367, 132973, doi:10.1016/j.jclepro.2022.132973.
  8. Li, J.; Ma, J.; Wei, W.Study of regional differences in energy carbon emission efficiency among eight comprehensive economic zones in China. The Journal of Quantitative & Technical Economics2020, 37, 109–129, doi: 10.13653/j.cnki.jqte.2020.06.006.
  9. Xin, L.; Sun, H.; Xia, X. Spatial–Temporal Differentiation and Dynamic Spatial Convergence of Inclusive Low-Carbon Development: Evidence from China. Environ Sci Pollut Res2023, 30, 5197–5215, doi:10.1007/s11356-022-22539-2.
  10. Zhou, W.; Cheng, F. Study on the willingness of inter-governmental cooperation of city clusters under the orientation of regional coordinated development - a textual analysis based on the government work reports of cities in three major city clusters. Urban Problems2022, 12–23, doi:10.13239/j.bjsshkxy.cswt.220702.

For specific revisions, please refer to the line 703-774 on page 21-22 of the Revised Manuscript (with changes marked).

Comments 5:

Policy recommendations should be formulated based on the findings of the study. The authors are requested to scrutinize this point carefully so that the policy recommendations and the research findings correspond to each other.

Response:

Thank you for your constructive comments on the policy recommendations. We respond as follows:

According to your suggestion, we have rewritten the third point of the policy recommendation to ensure that it is closely related to the research conclusion.

Revision

6.2 Policy implications

Thirdly, the local government should use land-space constraints on economic activities to promote a cleaner transformation of the industrial structure and the process of promoting green technologies. Transforming the existing unreasonable and unclean industrial structure is essential for developing an intensive and efficient economic growth mode. On the one hand, local governments need to protect patented technologies and promote enterprises' green technology research and development process. On the other hand, the government needs to promote the cleanliness of the energy structure and strengthen the rational allocation of industrial development and energy structure. In addition, the government should play an organizing and guiding role to promote green and healthy production methods such as recycling, high efficiency, and emission reduction and to improve the city's scientific and technological innovation capacity and technology spillover effect. In this way, a virtuous green circular economy can be formed within and among cities. 

For specific revisions, please refer to the line 841-853 on page 24 of the Revised Manuscript (with changes marked).

Reviewer 3 Report

Review of

 How does intensive land use affect low-carbon transition in China? New evidence from the spatial econometric analysis”

Manuscript ID: Land -2538684

The paper concerns a relationship between intensive land use (ILU) and low carbon transition (LCT) in China. The issue deals with one of the most important global environmental problems that is global warming. It is well-known that due to its rapid economic development China is the world's largest emitter of carbon dioxide. Therefore, there is an undoubted and significant need for various solutions that enable to reduce greenhouse gas emission and support the transition to a low-carbon economy.

There is a large number of current studies and articles on relationship between land-use intensity and low-carbon transitions in China. The main differences lie in the way the problem is formulated, what quantitative/qualitative indicators are taken into account, applied coefficients and finally – in statistical analysis that are applied. According to the Authors, they developed “a more comprehensive analysis from a spatiotemporal perspective”. Indeed, the methodological part of the study is very extensive. I cannot undertake to evaluate statistical methods and models that are applied but I can admit that their objectives, conditions and main stages are rather well presented.

Activities associated with implementation of low-carbon transition policy should be taken at different levels: national, regional and local. The paper presents rather general approach, suitable for the national or regional level. Conclusions and policy implications concern top-level (top-down?) actions to be taken by government and authorities of agglomerations or provinces. The integration of different activities is fundamental. This issues should be underlined in the abstract and in conclusions.

Some of the conclusions arise from the specific characteristics of China. I found it rather interesting (from European perspective) that the spillover effect of new urbanization exists in the range of 0-450 km and peaks at about 250 km.

The layout of the paper and the research design is correct. However, I have some additional comments.

Comment 1:  Keywords:

I suggest to add keywords: national and regional policy (spatial policy), urban spatial planning (?).

Comment 2:  Introduction 

I suggest to add a map with spatial distribution of current land use carbon emissions for provinces in China. It will enable to show carbon emissions from the quantitative and spatial point of view. Furthermore, it might facilitate the understanding results concerning spillover effects on the low-carbon transition for individual regions.

Comment 3:  Introduction  - lines 77-78

It is estimated that optimizing land use will contribute 27.6% to China's carbon emission reduction in 2020”. – since it is 2023 I suggest to change into “it was estimated” and add some comment whether these values are still valid.

Comment 4: Spatial heterogeneity - lines 577-578

Be more specific regarding city levels – define extra large, large, moderate and small cities. Refer also to 283 cities which are the subject of the study.

 Comment 5: Discussion - lines 661 – 668

I miss the spatial presentation of the results. Which are the eastern, central or  western regions when you talk about spill-over effects on the low-carbon transition of neighboring cities.   

Date 3ed.of  August 2023

Author Response

Response to Reviewer #3:

The paper concerns a relationship between intensive land use (ILU) and low carbon transition (LCT) in China. The issue deals with one of the most important global environmental problems that is global warming. It is well-known that due to its rapid economic development China is the world's largest emitter of carbon dioxide. Therefore, there is an undoubted and significant need for various solutions that enable to reduce greenhouse gas emission and support the transition to a low-carbon economy.

There is a large number of current studies and articles on relationship between land-use intensity and low-carbon transitions in China. The main differences lie in the way the problem is formulated, what quantitative/qualitative indicators are taken into account, applied coefficients and finally – in statistical analysis that are applied. According to the Authors, they developed “a more comprehensive analysis from a spatiotemporal perspective”. Indeed, the methodological part of the study is very extensive. I cannot undertake to evaluate statistical methods and models that are applied but I can admit that their objectives, conditions and main stages are rather well presented.

Activities associated with implementation of low-carbon transition policy should be taken at different levels: national, regional and local. The paper presents rather general approach, suitable for the national or regional level. Conclusions and policy implications concern top-level (top-down?) actions to be taken by government and authorities of agglomerations or provinces. The integration of different activities is fundamental. These issues should be underlined in the abstract and in conclusions.

Some of the conclusions arise from the specific characteristics of China. I found it rather interesting (from European perspective) that the spillover effect of new urbanization exists in the range of 0-450 km and peaks at about 250 km.

The layout of the paper and the research design is correct. However, I have some additional comments.

Response:

We would like to thank the reviewer for your careful review and comments on our manuscript. Your comments help us improve the quality of our work. In your comments, you pointed out accurately the existing shortcomings we ignored, and also gave the practical solutions. We are honored to receive your approval of the value of this manuscript. We fully agree and have made revisions according to each of your comments. Our responses are organized point by point based on your comments.

Comments 1:

Keywords:

I suggest to add keywords: national and regional policy (spatial policy), urban spatial planning (?)

Response:

Thank you for your constructive comments on keywords. We have reextracted the keywords and added the keywords you suggested. The revised sections are as follows:

Revision

Keywords: intensive land use; low carbon transformation; Industrial structure transformation; technology spillovers; national and regional policy; land space planning

For specific revisions, please refer to the line 27-28 on page 1 of the Revised Manuscript (with changes marked).

Comments 2:

Introduction:

I suggest to add a map with spatial distribution of current land use carbon emissions for provinces in China. It will enable to show carbon emissions from the quantitative and spatial point of view. Furthermore, it might facilitate the understanding results concerning spillover effects on the low-carbon transition for individual regions.

Response:

Thank you for your valuable comments and suggestions on the Introduction. Your suggestions help improve this manuscript's readability and make it more informative. We accept your suggestion and add a map with the spatial distribution of current land use carbon emissions for provinces in China in the introduction section. Meanwhile, we describe the content of the map in the introduction section. We expect that it will enable readers to clearly see the regional spatial differences. The modifications are as follows:

Revision

1.Introduction

…The loss of ecosystem carbon stocks will be exacerbated by unintentional land expansion, and the overconcentration of human activities brought about by expansion will also generate high consumption and emissions [15]. As the world's largest carbon emitter, land use carbon emissions have become an essential source of carbon emissions in China, reaching 3.2 × 109t in 2015, an increase of about 2.45 times compared with 1999. As of 2020, China's land-use carbon emissions will remain high (see Figure 1). Assuming that the 1.5°C global temperature control target of the Paris Agreement is to be achieved, further attention needs to be paid to the critical role of intensive land use in the low-carbon transition of the economy. China is...

In China, fiscal revenue from land concessions has long been local governments' primary income source. To solve the fiscal balance gap, some local governments have been keen to attract industrial investment by taking advantage of their resource endowment and geographic location [18], dramatically expanding industrial scale while promoting development. This process has not only resulted in massive waste of urban land and rapid urbanization but also led to differences in the spatial pattern of land use carbon emissions [19]. With China's coordinated economic development strategy deepening, the flow of technology, personnel, capital, and other factors between regions has further accelerated. The spatial correlation of economic development, energy consumption, and agricultural activities has broken through the limitations of geographic location. The spatial correlation of land-use carbon emissions will also be further complicated. Under the spatial differences in population distribution and economic resources, the differences in land carbon emissions of each province in China are apparent, carbon emissions are high in the eastern coastal areas (Figure 1). In this context, it is significant to carry out a study on the spatial differences in carbon emissions from land use for the synergistic emission re-duction of regional land use.

Figure 1..Land use carbon emissions for 31 provinces in China in 2020

For specific revisions, please refer to the line 58-96 on page 2-3 of the Revised Manuscript (with changes marked).

Comments 3:

Introduction  - lines 77-78

It is estimated that optimizing land use will contribute 27.6% to China's carbon emission reduction in 2020”. – since it is 2023 I suggest to change into “it was estimated” and add some comment whether these values are still valid.

Response:

Thank you for your valuable comments and suggestions. We have made an ambiguity in describing the results of this research. We apologize for the disturbance caused to you. We have re-summarized and rewritten the sentences. The modifications are as follows:

Revision

1. Introduction

…The regulation encourages small-scale centralized and intensive land use and emphasiz-es green and livable land use [16]. Huang and Lai (2021) assessed the carbon emission reduction contribution of the Outline of China's Overall Land Use Plan (2005-2020). Based on 2005, optimizing the land use structure will contribute 27.6% to the achievement of the target of carbon emission reduction of 40% to 45% per unit of GDP in 2020 [17]. In the context of the country's emphasis on coordinated "economic-ecological" development, intensive land use is cru-cial for China to achieve the goal of "carbon neutrality" and low-carbon transition.

For specific revisions, please refer to the line72-76 on page 2 of the Revised Manuscript (with changes marked).

Comments 4:

Spatial heterogeneity - lines 577-578

Be more specific regarding city levels – define extra-large, large, moderate and small cities. Refer also to 283 cities which are the subject of the study.

Response:

Thank you for your valuable comments and suggestion. Based on your suggestion, in the paragraph analyzing city-level heterogeneity, we describe in detail the classification criteria and indicators for the four types of cities. At the same time, we report the sample size of the four types of cities. The modifications are as follows:

Revision

4.5.2. Spatial heterogeneity

Cities at different size. We use the year-end population of the city district as a proxy variable for city size. Given the frequent changes in the administrative divisions of city districts in many cities, we use the city size division criteria published by China in 2014 to select cities. Based on the total year-end population of city municipal districts, the 283 city samples can be categorized into four groups: extra-large cities (more than 5 million people), large cities (1 million to 5 million people), moderate cities (half a million to 1 million people), and small cities (less than half a million people). Among all the cities in the sample, there are 13 extra-large cities, 127 large cities, 98 moderate cities, and 45 small cities. As shown in Panel B of Table 9: The indirect effects for medium-sized and large cities are 7.954 and 5.836…

For specific revisions, please refer to the line665-672 on page 20 of the Revised Manuscript (with changes marked).

Comments 5:

Discussion - lines 661 – 668

I miss the spatial presentation of the results. Which are the eastern, central or western regions when you talk about spill-over effects on the low-carbon transition of neighboring cities.

Response:

Thank you for your valuable comments and suggestion. We have rewritten this paragraph in the discussion. Specifically, we point out the importance of spatial differences in ILU policies in the context of the map of carbon emissions from land use in China's provinces that we added in the previous section (as you suggested), and taking into account the specific economic development of cities in the east, central or western regions. The modifications are as follows:

Revision

5.Discussion

To promote China's low-carbon transition, paying attention to the spatial differences and regional cooperation in the environmental benefits of ILU policies is essential. This is consistent with the findings of current research in other countries [60][61]. First, it is necessary to pay attention to geographic location differences. The results of the heterogeneity regression show that only the cities in the eastern region can implement ILU policies while generating positive spillover effects on the low-carbon transition of neighboring cities, and the results in the central and western regions are not significant. In fact, the land carbon emissions of the eastern region, including Beijing, Tianjin, Shanghai, Jiangsu Province, Zhejiang Province, etc., account for a high proportion of the national emissions (see Figure 1); however, eco-efficiency and energy use efficiency are also higher in the east-ern coastal region than in the central and western regions [62][63]. Taken together, although the eastern region faces more substantial pressure to reduce emissions, it has developed a more inclusive green land use system over the years and has experience in urban spatial planning and low-carbon environmental management, thus creating a demonstration effect on neighboring cities. These achievements may be related to the greater concentration of talent, technology, and innovation in the eastern region, which could receive further attention. On the contrary, the central and western regions, such as Sichuan, Hubei, Henan, and other provinces, are all in rapid economic development, with large populations and high pressure on land use carbon emissions. Promoting ILU in cities in the central and western regions is more challenging and requires more advanced experience from the eastern regions. Secondly, it is…

For specific revisions, please refer to the line746-766 on page 22 of the Revised Manuscript (with changes marked).

Reviewer 4 Report

The authors propose an analysis aimed at highlighting the relationship between intensive land use (ILU) and low-carbon transition (LCT), considering direct and spatial effects.

 The paper is very interesting and overall well organized.

It is suggested to:

- Integrate a table with all acronyms;

- Characterize models 1,2,3 and 4 in Table 6;

- Characterize patterns 1 and 2 in Table 7.

-Integrate in the Discussion section some comparisons with the international literature

It is recommended to review the text in English

Author Response

Response to Reviewer #4:

The authors propose an analysis aimed at highlighting the relationship between intensive land use (ILU) and low-carbon transition (LCT), considering direct and spatial effects.

 The paper is very interesting and overall well organized.

Response:

We would like to thank the reviewer for your careful review and comments on our manuscript. Your comments help us improve the quality of our work. In your comments, you pointed out accurately the existing shortcomings we ignored, and also gave the practical solutions. We are honored to receive your approval of the value of this manuscript. We fully agree and have made revisions according to each of your comments. Our responses are organized point by point based on your comments.

Comments 1:

Integrate a table with all acronyms

Response:

Thank you for your valuable comments and suggestion. According to your suggestion, we have added a table with all acronyms in the appendix (Appendix A.1) to improve the readability of the whole text. The revised sections are as follows:

Revision

Appendix

A.1. Summary table of acronyms

Acronyms

Description

LCT

Low-carbon transition

ILU

Intensive land use

ER

Environmental regulation

TEM

annual average temperature

OPEN

Openness to foreign investment

GOV

Government Intervention

AGG

Industrial agglomeration

MAK

Marketization

FIAN

Financial development

IS

Industrial structure transformation

TS

Technology spillover

For specific revisions, please refer to the line 1029-1030 on page 27 of the Revised Manuscript (with changes marked).

Comments 2:

Characterize models 1,2,3 and 4 in Table 6

Response:

Thank you for your valuable comments and suggestions. We further analyze the characteristics of the four models in Table 6 in Subsection 4.2. Since the scope of this revision involves the order of the tables, Table 6 in the original manuscript is Table 5 in the revised manuscript, and we hope you will understand this when reading the revised manuscript. The revised sections are as follows:

Revision

4.2. Robustness tests

We first conduct a parallel trend test to analyze whether policy evaluation can be conducted using the double difference approach. After the results showed that this important test was passed, we proceeded to model estimation. Table 5 reports the regression results of the spatial difference-in-differences model for the four spatial weight matrices (regression results of equation (10)). As we can see, the indirect effects of …

In particular, there are some characteristics of Models 1-4 results based on spatial weights of different geographical elements. As can be seen, the coefficient of distance spatial weights (Wdis, 0.342) and the coefficient of economic distance spatial weights (Wecondis, 0.343) are higher than the coefficient of indirect effect of economic spatial weights (Wecon, 0.112) and higher than the coefficient of neighboring spatial weights (Wadj,0.051). It may be because the pilot cities have made more efforts to promote low-carbon, green, inclusive, and smart cities. The demonstration effect and economic correlation effect on the associated cities reflect the positive impact of the low-carbon transition of the cities. In addition, the magnitude of the coefficient suggests that this demonstration effect is more likely to be constrained by geographical distance.

For specific revisions, please refer to the line 500-513 on page 15 of the Revised Manuscript (with changes marked).

Comments 3:

Characterize patterns 1 and 2 in Table 7.

Response:

Thank you for your valuable comments and suggestions. We further analyze the characteristics of models 1-2 in Table 7 of the original manuscript. Table 7 in the original manuscript is Table 6 in the revised manuscript, and we hope you will understand this when reading the revised manuscript. The revised sections are as follows:

Revision

4.3.1. Channel mechanism of industrial structure transformation

Model 1 in Table 6 shows the impact of intensive land use on industrial structure upgrading, and it can be seen that the impact is positive and significant, indicating that ILU promotes China's low-carbon transformation through the IS influence mechanism. Specifically, ILU, as a long-term national policy, will constrain the disorderly expansion of enterprise land use and promote the economization of enterprise production and operation in the coming period. In addition, the objective constraints of ILU on urban space will limit the entry of highly polluting and low-value-added industries. Therefore, in cities with intensive land utilization, industries can obtain green development and regulate the industrial layout by regulating the proportion of clean industries, thus promoting low-carbon development.

4.3.2. Channel mechanism of technology spillover

As shown in Model 2 in Table 6, the effect of intensive land use on technology spillovers is significant, indicating that ILU promotes China's low-carbon transition through the influence mechanism of TS. Specifically, ILU's land use restrictions on enterprises can first force enterprises to increase investment and research in green products and new materials. Second, compared with the standardized and large-scale production of the secondary industry, the knowledge and technology-intensive tertiary sector tends to have higher value-added, lower energy consumption and is more in line with the need for in-tensive land use. It means cities with intensive land utilization have built a good platform for technology R&D and dissemination. Third, the positive psychological effect of ILU on low-carbon development of industries should not be ignored. Positive public opinion encourages regional industrial enterprises and regional enterprises to imitate each other and technological innovation, thus promoting the low-carbon transformation of the region.

For specific revisions, please refer to the line 524-561 on page 16-17 of the Revised Manuscript (with changes marked).

Comments 4:

Integrate in the Discussion section some comparisons with the international literature

Response:

Thank you for your constructive comments on the discussion. According to your suggestion, we compare it with the existing literature to emphasize the feasibility of the discussion of this paper. The revised sections and literature added is shown below:

Revision

5.Discussion

…The baseline regression results in this paper verify that ILU has a significant positive spillover effect on low-carbon transformation, providing ideas for promoting ILU development to realize low-carbon urban transformation and carbon neutrality. This result is consistent with the findings of Shang et al. (2022) [56], but we go a step further by considering the in-fluence of spatial factors and drawing conclusions about spillover effects and spatial boundaries. In addition, our findings further confirm the greenness and sustainability of China's ILU policy and urban spatial optimization [57]. China has implemented and is implementing integrated land use policies (e.g., Provisions on Saving and intensive Land Use (2014) [16]. As the spatial mainstay of …

…Compared to ecological carbon sink improvement and land use efficiency, the increase in land use density cannot significantly promote the local low-carbon transition. This result is similar to that of Albert et al. (2015) [58] based on data from European countries. Considering that land use density is related to the city's actual built-up area and the population's carrying capacity. This situation may arise because the current land intensification in China is still in the stage of capital intensification [59]. The most important vehicle for urban development is…

To promote China's low-carbon transition, paying attention to the spatial differences and regional cooperation in the environmental benefits of ILU policies is essential. This is consistent with the findings of current research in other countries [60][61]. First, it is necessary to pay attention to geographic location differences. The results of …

The literature added is shown below:

  1. Shang, Y.; Xu, J.; Zhao, X. Urban Intensive Land Use and Enterprise Emission Reduction: New Micro-Evidence from China towards COP26 Targets. Resources Policy2022, 79, 103158, doi:10.1016/j.resourpol.2022.103158.
  2. Li, X.; Wang, L. Does Administrative Division Adjustment Promote Low-Carbon City Development? Empirical Evidence from the “Revoke County to Urban District” in China. Environ Sci Pollut Res2023, 30, 11542–11561, doi:10.1007/s11356-022-22653-1.
  3. Baur, A.H.; Förster, M.; Kleinschmit, B. The Spatial Dimension of Urban Greenhouse Gas Emissions: Analyzing the Influence of Spatial Structures and LULC Patterns in European Cities. Landscape Ecol2015, 30, 1195–1205, doi:10.1007/s10980-015-0169-5.
  4. Wu, Y.; Ren, Y.; Xu, Z. Market allocation of land elements under the national spatial planning system: theory, mechanism and model. China Land Sciences2023, 37, 28–37.
  5. Bridge, G.; Bouzarovski, S.; Bradshaw, M.; Eyre, N. Geographies of Energy Transition: Space, Place and the Low-Carbon Economy. Energy Policy2013, 53, 331–340, doi:10.1016/j.enpol.2012.10.066.
  6. Hill, D. Regional Cooperation and Asia’s Low Carbon Economy Transition: The Case of New Zealand. In Investing on Low-Carbon Energy Systems: Implications for Regional Economic Cooperation; Anbumozhi, V., Kalirajan, K., Kimura, F., Yao, X., Eds.; Springer: Singapore, 2016; pp. 309–326 ISBN 978-981-10-0761-3.

For specific revisions, please refer to the line 703-748 on page 21-22 of the Revised Manuscript (with changes marked).

Comments 5:

Comments on the Quality of English Language: It is recommended to review the text in English

Response:

Thanks for your suggestions, and your point is well-taken. We apologize for the problems with the language. The language of the paper has been carefully checked and corrected with the assistance of native English-speaking experts. We hope that language fluency has been substantially improved. If our revision is not up to your standard yet, we would be happy to revise it again.

Reviewer 5 Report

1. This manuscript examines the effect of the significance of intensive land use (optimizing land use) on China's slow-carbon transition and deals with important themes. China's carbon emissions are not small, and we hope that it will have an effect on policy. 2. Nature positive is oriented in every country, and promotion of "economic-ecological" is important. Therefore, "2. Heterogeneity Discussion" and "4. Mediating Test" in the introduction chart are important processes. I hope that the authors will continue to carefully present the examination methods and results in this process. 3. No paper is perfect and every paper has its flaws. These authors are credible in their discussion while understanding them. It is a natural conclusion, and it can be said that it is mediocre, but it is meaningful to verify and present it. 4. This is described in relation to above 3 . The conclusion is too simplistic. (1) What is the difference between a case (region) where ILU contributes to low carbonization and a case (region) where it does not? (2) What kind of industrial transfers and technological spillovers are effective for ILU, which contributes to low carbonization? I would like you to add such considerations.

Author Response

Response to Reviewer #5:

1. This manuscript examines the effect of the significance of intensive land use (optimizing land use) on China's slow-carbon transition and deals with important themes. China's carbon emissions are not small, and we hope that it will have an effect on policy.

Response:

We would like to thank the reviewer for your careful review and comments on our manuscript. Your comments help us improve the quality of our work. In your comments, you pointed out accurately the existing shortcomings we ignored, and also gave the practical solutions. We are honored to receive your approval of the value of this manuscript. We fully agree and have made revisions according to each of your comments. Our responses are organized point by point based on your comments.

2. Nature positive is oriented in every country, and promotion of "economic-ecological" is important. Therefore, "2. Heterogeneity Discussion" and "4. Mediating Test" in the introduction chart are important processes. I hope that the authors will continue to carefully present the examination methods and results in this process.

Response:

Thank you for your constructive comments on the Heterogeneity and Mediating Test.

As it is difficult to add more details in the introduction chart, following your suggestion, we further described the mechanism variables' measurement in Section 3.3.4 and added theoretical justifications and literature support. Second, we added the explanation of the sample division criteria in the Heterogeneity Test (Section 4.5.2). The revised sections are shown below:

Revision

3.3.4. Other variables

2. Channel variables

Based on the theoretical analysis in the previous subsections, the two key mechanism variables for channel analysis are industrial structure transformation (IS) and technology spillover (TS).

Industrial structure transformation (IS). The upgrading of industrial structure towards cleanliness is the key to realizing the goal of green development. The current indicators for industrial structure upgrading mainly use internal structure change, energy consumption per unit GDP of industry, and product sales of pollution-intensive industries. We utilize the entropy value method to determine the degree of cleaner transformation of industrial structure. In this paper, we refer to Zhang et al. (2023) to construct the indicator system from two aspects of clean energy consumption and clean production [31]. Clean energy consumption is measured by the ratio of total industrial energy consumption to industrial added value; clean production is expressed by the ratio of regional industrial added value to carbon emissions. Through the dimensionless quantization of the indicators, the entropy value method is then used to identify the degree of cleaner transformation of the industrial structure.

Technology spillover (TS). Generally speaking, due to China's imperfect patent guarantee mechanism and relatively backward R&D capability, it is difficult for enterprises to convert R&D inputs into green innovation outputs. In contrast, the number of green patents can reflect the actual innovation outputs more objectively. We use the number of green patent acquisitions obtained in one year as a proxy variable for technology spillovers.

4.5.2. Spatial heterogeneity

Cities with different geographic locations. As for the natural location of the province, different regions have distinct economic development goals, land use regulations, and contaminant pressure. Accordingly, the im-pact of IER may be affected by the geographical location, so we have divided the sample into three sample subgroups including eastern, western, and central regions. The results are shown in Panel A of Table 9. In the eastern region, the spillover effect of ILU on urban low-carbon transition is…

Cities at different size. We use the year-end population of the city district as a proxy variable for city size. Given the frequent changes in the administrative divisions of city districts in many cities, we use the city size division criteria published by China in 2014 to select cities. Based on the total year-end population of city municipal districts, the 283 city samples can be categorized into four groups: extra-large cities (more than 5 million people), large cities (1 million to 5 million people), moderate cities (half a million to 1 million people), and small cities (less than half a million people). Among all the cities in the sample, there are 13 extra-large cities, 127 large cities, 98 moderate cities, and 45 small cities. As shown in Panel B of Table 9…

For specific revisions, please refer to the line 408-425 on page 11-12, the line 653-672 on page 20 of the Revised Manuscript (with changes marked).

3. No paper is perfect and every paper has its flaws. These authors are credible in their discussion while understanding them. It is a natural conclusion, and it can be said that it is mediocre, but it is meaningful to verify and present it.

4. This is described in relation to above 3. The conclusion is too simplistic.

(1) What is the difference between a case (region) where ILU contributes to low carbonization and a case (region) where it does not?

(2) What kind of industrial transfers and technological spillovers are effective for ILU, which contributes to low carbonization? I would like you to add such considerations.

Response:

Thank you very much for your kind and constructive comments. We have done our best to revise the manuscript, but if any additional revision is needed, we will certainly do so under your directions. We respond as follows:

For question (1): we interpret the heterogeneity in further detail in Discussion (Section 5), taking into account the previous map of the spatial distribution of carbon emissions from land use in China's provinces (Figure 1 in Introduction) and the actual situation of China's regional development, and add comparisons with other literature. The revised sections are as follows:

Revision

5.Discussion

…Second, it is essential to emphasize the means of ILU policies. Compared to ecological carbon sink improvement and land use efficiency, the increase in land use density cannot significantly promote the local low-carbon transition. This result is similar to that of Albert et al. (2015) [58] based on data from European countries. Considering that land use density is related to the city's actual built-up area and the population's carrying capacity. This situation may arise because the current land intensification in China is still in the stage of capital intensification [59]. The most important vehicle for urban development is the construction land, and capital investment is concentrated in construction land. In this case, an increase in land use density will increase infrastructure and energy investment. Therefore, decisions about urban folding, spatial planning, and urbanization development need to be implemented prudently.

To promote China's low-carbon transition, paying attention to the spatial differences and regional cooperation in the environmental benefits of ILU policies is essential. This is consistent with the findings of current research in other countries [60][61]. First, it is necessary to pay attention to geographic location differences. The results of the heterogeneity regression show that only the cities in the eastern region can implement ILU policies while generating positive spillover effects on the low-carbon transition of neighboring cities, and the results in the central and western regions are not significant. In fact, the land carbon emissions of the eastern region, including Beijing, Tianjin, Shanghai, Jiangsu Province, Zhejiang Province, etc., account for a high proportion of the national emissions (see Figure 1); however, eco-efficiency and energy use efficiency are also higher in the east-ern coastal region than in the central and western regions [62][63]. Taken together, although the eastern region faces more substantial pressure to reduce emissions, it has developed a more inclusive green land use system over the years and has experience in urban spatial planning and low-carbon environmental management, thus creating a demonstration effect on neighboring cities. These achievements may be related to the greater concentration of talent, technology, and innovation in the eastern region, which could receive further attention. On the contrary, the central and western regions, such as Sichuan, Hubei, Henan, and other provinces, are all in rapid economic development, with large populations and high pressure on land use carbon emissions. Promoting ILU in cities in the central and western regions is more challenging and requires more advanced experience from the eastern regions. Secondly, it is necessary to pay attention to urban level differences. The positive spillover effect of ILU on urban low-carbon transition is not evident in small and medium-sized cities and urban circles. These findings reflect two aspects: small and medium-sized cities have weaker governance capacity and may face more difficult ecological governance and spatial layout adjustment [57]. The second is that administrative barriers within the city-region have not yet been broken down [64], and there is less willingness to cooperate between city clusters [65], which limits the direct and spillover effects of ILU policies in the city-region on the low-carbon transition. The above conclusions provide ideas for future synergistic promotion of low-carbon transformation in Chinese cities.

For question (2), in the mechanism identification section (Section 4.3), we explain the mechanism formation process in further detail by combining the empirical results and the theoretical analysis in the previous section. The revised sections are as follows:

Revision

4.3.1 Channel mechanism of industrial structure transformation

Model 1 in Table 6 shows the impact of intensive land use on industrial structure upgrading, and it can be seen that the impact is positive and significant, indicating that ILU promotes China's low-carbon transformation through the IS influence mechanism. Specifically, ILU, as a long-term national policy, will constrain the disorderly expansion of enterprise land use and promote the economization of enterprise production and operation in the coming period. In addition, the objective constraints of ILU on urban space will limit the entry of highly polluting and low-value-added industries. Therefore, in cities with intensive land utilization, industries can obtain green development and regulate the industrial layout by regulating the proportion of clean industries, thus promoting low-carbon development.

4.3.2 Channel mechanism of technology spillover

As shown in Model 2 in Table 6, the effect of intensive land use on technology spillovers is significant, indicating that ILU promotes China's low-carbon transition through the influence mechanism of TS. Specifically, ILU's land use restrictions on enterprises can first force enterprises to increase investment and research in green products and new materials. Second, compared with the standardized and large-scale production of the secondary industry, the knowledge and technology-intensive tertiary sector tends to have higher value-added, lower energy consumption and is more in line with the need for intensive land use. It means cities with intensive land utilization have built a good platform for technology R&D and dissemination. Third, the positive psychological effect of ILU on low-carbon development of industries should not be ignored. Positive public opinion encourages regional industrial enterprises and regional enterprises to imitate each other and technological innovation, thus promoting the low-carbon transformation of the region.

For specific revisions, please refer to the line 734-774 on page 21-22, the line 524-570 on page 16-17 of the Revised Manuscript (with changes marked).